# The risk factors associated with treatment-related mortality in 16,073 kidney transplantation—A nationwide cohort study

**Hyunji Choi**[1]°, **Woonhyoung Lee**[1]°, **Ho Sup Lee**[2], **Seom Gim Kong**[3], **Da Jung Kim**[2], **Sangjin Lee**[4], **Haeun Oh**[1], **Ye Na Kim**[5], **Soyoung Ock**[6], **Taeyun Kim**[6], **Min-Jeong Park**[7], **Wonkeun Song**[7], **John Hoon Rim**[8,9], **Jong-Han Lee**[10], **Seri Jeong**[7] *

1 Department of Laboratory Medicine, Kosin University College of Medicine, Busan, South Korea,
2 Department of Hematology-Oncology, Kosin University College of Medicine, Busan, South Korea,
3 Department of Pediatrics, Kosin University College of Medicine, Busan, South Korea, 4 Graduate School, Department of Statistics, Pusan National University, Busan, South Korea, 5 Department of Nephrology, Kosin University College of Medicine, Busan, South Korea, 6 Department of Internal Medicine, Kosin University College of Medicine, Busan, South Korea, 7 Department of Laboratory Medicine, Hallym University College of Medicine, Seoul, South Korea, 8 Department of Pharmacology, Yonsei University College of Medicine, Seoul, South Korea, 9 Department of Medicine, Physician-Scientist Program, Yonsei University Graduate School of Medicine, Seoul, South Korea, 10 Department of Laboratory Medicine, Yonsei University Wonju College of Medicine, Wonju, South Korea

☯ These authors contributed equally to this work.
* hehebox@naver.com

**Data Availability Statement:** The repository data for public release is not available because of the personally identifiable information. The full dataset includes clinic centers in which they attend,

## Abstract

Mortality at an early stage after kidney transplantation is a catastrophic event. Treatment-related mortality (TRM) within 1 or 3 months after kidney transplantation has been seldom reported. We designed a retrospective observational cohort study using a national population-based database, which included information about all kidney recipients between 2003 and 2016. A total of 16,073 patients who underwent kidney transplantation were included. The mortality rates 1 month (early TRM) and 3 months (TRM) after transplantation were 0.5% (n = 74) and 1.0% (n = 160), respectively. Based on a multivariate analysis, older age (hazard ratio [HR] = 1.06; $P < 0.001$), coronary artery disease (HR = 3.02; $P = 0.002$), and hemodialysis compared with pre-emptive kidney transplantation (HR = 2.53; $P = 0.046$) were the risk factors for early TRM. Older age (HR = 1.07; $P < 0.001$), coronary artery disease (HR = 2.88; $P < 0.001$), and hemodialysis (HR = 2.35; $P = 0.004$) were the common independent risk factors for TRM. In contrast, cardiac arrhythmia (HR = 1.98; $P = 0.027$) was associated only with early TRM, and fungal infection (HR = 2.61; $P < 0.001$), and epoch of transplantation (HR = 0.34; $P < 0.001$) were the factors associated with only TRM. The identified risk factors should be considered in patient counselling, selection, and management to prevent TRM.

## Introduction

After kidney transplantation, patients with end-stage renal disease (ESRD) had better survival, improved cognition, and less economic burden than those who continued with dialysis [1–3].

insurance conditions. Therefore, concerning privacy risks, the data is managed by authorized executive supervisor. If one researcher asks to access data, the person in charge releases data with blind identification for the discrete requirements and the data should be analyzed only in permitted rooms in centers of National Health Insurance Service. Subsets of data limited to anonymisable information obtained and analyzed during this study are included in this article (tables, figures, and supporting information). Contact information for a data access committee is listed as follows: National Health Insurance Sharing Service, Tel: 82-33-736-2432; Official internet site: https://nhiss.nhis.or.kr/bd/ay/bdaya001iv.do. Other researchers can access these data in the same manner as the authors and the authors did not have any special access privileges.

**Funding:** This work was supported by the National Research Foundation of Korea (NRF) grant, funded by the Korean government (Ministry of Science and ICT) [NRF-2017R1C1B2004597] (to SJ). URL: http://www.nrf.re.kr/index The funders had no role in study design, data collection and analysis, decision to publish, or preparation of the manuscript.

**Competing interests:** The authors have declared that no competing interests exist.

**Abbreviations:** CAD, coronary artery disease; CI, confidence interval; CMV, cytomegalovirus; ESRD, end-stage renal disease; HIRA, Health Insurance Review and Assessment Service; HR, hazard ratio; ICD-10-CM, International Classification of Disease, 10th revision, Clinical Modification; NA, not applicable; NHI, National Health Insurance; RID, Rare Intractable Disease; TRM, treatment-related mortality.

Kidney transplantation has improved over the past decades [4]. However, some kidney recipients still die at an early stage after surgery, which is catastrophic for both the patient and medical staff.

Investigation of treatment-related mortality (TRM), which is a concept different from disease-related mortality, is important for improved survival after treatment. It provides information about factors that require intensive care and medical decisions during critical period [5,6]. In cardiovascular procedures or major abdominal surgery, 30-day mortality after surgery is considered TRM [7–9]. In addition, 90-day postoperative mortality is a legitimate measure of hepatobiliary–pancreatic surgery [10]. Furthermore, 90-day mortality rate is a good predictor of postoperative index in the field of hepatectomy, colectomy, and pneumonectomy [10–13]. Data about 1-year mortality after kidney transplantation or long-term outcome were well reported [14–17]. Most reports have shown the results of kidney transplantation after 1 [18], 5 [16], and greater than 10 years [19]; however, studies about 1- or 3-month mortality were extremely limited [20,21].

The present study was based on the use of a comprehensive database, which is operated by the National Health Insurance (NHI) of the Korean government. This database contains all the records of healthcare utilization among inpatients and outpatients particularly kidney recipients who were enrolled in the Rare Intractable Disease (RID) system and who received additional medical financial support. The registration is confirmed by a certified physician based on the RID criteria, which reflect international guidelines. Therefore, the use of this database was suitable for the investigation of TRM among kidney recipients.

Using this database, we performed a comprehensive population-based analysis to investigate the risk factors and causes of TRM after kidney transplantation. It would facilitate pre- and post-transplantation assessment and management, which contributed to the improvement of the survival of kidney recipients.

# Materials and methods

## Study design

This was a retrospective and observational cohort study that used prospectively registered national data sets for reimbursement purposes. All patients who underwent kidney transplantation procedures (Z94.0 code of the International Classification of Disease, 10th revision, Clinical Modification [ICD-10-CM]) at any Korean medical center from January 2003 to December 2016 were included. We defined death within 1 and 3 months after kidney transplantation as early TRM and TRM, respectively. We investigated the risk factors related to early TRM and TRM and the causes of death.

## Ethics statement

This study was approved by the independent institutional review board of Kosin University Gospel Hospital (KUGH 2017-12-009) and was conducted in accordance with the Declaration of Helsinki. Moreover, the need for informed consent was waived because anonymity of personal information was maintained.

## Study population (patient selection)

The study included all patients who have been listed for kidney transplantation from January 2003 to December 2016 in the Health Insurance Review and Assessment Service (HIRA). The patients were registered in the HIRA database after kidney transplantation, as defined by the ICD-10-CM code Z94.0. During this period, 18,822 patients were enrolled in the database. We

excluded 2,726 patients who did not have complete demographic information and 59 patients who concurrently underwent other organ transplantations. The final cohort consisted of 16,037 patients. The records of medical visits, demographic characteristics, and death status were collected from the HIRA database for all kidney recipients.

## Study variables

We collected the following demographic data and baseline characteristics of kidney recipients from the HIRA database: age, sex, medical comorbidities focusing on cardiac and cerebrovascular diseases reported to be important causes of early mortality [16], dialysis status, cytomegalovirus (CMV) and fungal infection, and year of transplantation (S1 Table). The induction regimens such as basiliximab, and anti-thymocyte globulin were also extracted. CMV infection included CMV diseases (mononucleosis, pneumonitis, and hepatitis) and the post-transplant administration of antiviral agent (ganciclovir or valganciclovir) [22]. The ICD-10-CM codes for CMV disease were B27.1, B25.0, B25.1, B25.8, and B25.9. Fungal infection encompassed candidiasis and aspergillosis and post-transplant administration of antifungal agents (amphotericin, caspofungin, itraconazole, voriconazole, fluconazole, posaconazole, anidulafungin, and micafungin) [23].

## Data source

The data used in this study were obtained from the HIRA database, which is based on the NHI system operated by the Korean government. Healthcare institutions submit the medical data of all inpatients and outpatients in electronic format to the HIRA for reimbursement purposes. The claims data integrated by HIRA include all healthcare utilization information on inpatients and outpatients. Data about the demographic characteristics of the patients, principal diagnosis, comorbidities, prescription history, and performed procedures based on ICD-10-CM codes are included in this database. In this study, we obtained all data about kidney recipients from the RID program of the HIRA database registered between January 2003 and the end of December 2016. The Korean government assigned kidney transplantation to the RID system for reducing the payments of the patients. The diagnosis must be reviewed by the corresponding healthcare institution before submission to the NHI. Therefore, the data registered in the RID registry are verified and reliable.

The data for dialysis vintage, and donor state omitting in HIRA database were obtained from another database operated by the Korean Network for Organ Sharing system. In this database, the records of recipients who underwent kidney transplantation in 40 medical centers around the country were registered.

## Statistical analysis

We evaluated the TRM, risk factors, and causes of death of kidney recipients in Korea from 2003 to 2016. Descriptive statistics were used for patient characteristics correlated to early TRM and TRM. Comparisons of nominal and continuous variables between groups were assessed using chi-square test and Mann–Whitney U test, respectively. The median and interquartile range were used for non-normally distributed variables. Multivariate Cox proportional-hazards regression models adjusting age, sex, cardiac and cerebrovascular diseases, hemodialysis, infection, and epoch of transplantation were used to examine the variables correlated to TRM.

Statistical analyses were performed using the R statistical software (version 3.4.4; R Foundation for Statistical Computing, Vienna, Austria) and SAS statistical analysis software (version

9.4; SAS Institute Inc., Cary, NC, the USA). The two-tailed *P* values less than 0.05 were considered statistically significant.

## Results

### Characteristics of patients

A total of 16,073 patients who underwent kidney transplantation between 2003 and 2016 were included in our study cohort. The baseline characteristics of these patients are presented in Table 1. The median age of the patients was 47.0 years (1st to 3rd quartile range: 38.0–55.0 years). Our cohort consisted of 9,495 men and 6,578 women. Most patients received kidney from living donor (62.2%), followed by deceased (37.5%) and non-heart beating (0.3%) donors. The most common underlying disease was coronary artery disease (CAD) or cardiac arrhythmia, present in 10.3% of included patients. Most of patients received kidney transplantation after hemodialysis (82.1%). Regarding to induction therapy, basiliximab, and anti-thymocyte globulin were administered to 79.0%, and 11.4% of recipients, respectively. Cytomegalovirus (CMV) and fungal infections were more commonly reported at 3-month than 1-month (4.3% to 12.1% for CMV; 4.0% to 7.7% for fungus). The number of transplantation cases more than doubled from 2003–2009 (4,661 transplantations, 29.0% of included patients) to 2010–2016 (11,412 transplantations, 71.0% of included patients).

### Treatment-related mortality

Of the 16,073 patients, 74 (0.5%) and 160 (1.0%) died within 1 and 3 months after kidney transplantation, respectively. The overall cumulative incidence of mortality is shown in Fig 1A. The characteristics of kidney recipients who died within 1 and 3 months were compared to those of living patients, and such characteristics are summarized in Table 1. Based on this comparative analysis, the values of both early TRM and TRM rates significantly increased as the age group increased. In particular, the number of patients who died 1 month (6.8%) and 3 months (5.6%) after transplantation was five times higher than that of living patients (0.9%) aged over 70 years. The rates of recipients who died 1 month (n = 1, 1.4% for living; n = 2, 2.7% for deceased; and n = 2, 2.7% for non-heart beating) and 3 months (n = 5, 3.1% for living; n = 9, 5.6% for deceased; and n = 5, 3.1% for non-heart beating) after transplantation showed significant difference according to the donor state (*P* < 0.001). The number of patients with a history of cardiac disease, including coronary artery disease (CAD) (*P* < 0.001) and cardiac arrhythmia (*P* = 0.002), was significantly higher in the TRM groups than in the non-TRM groups. The recipients with TRM more frequently had undergone hemodialysis (*P* = 0.012 for early TRM; *P* = 0.001 for TRM). Patients with anti-thymocyte globulin showed significant relation to TRM (*P* < 0.001), whereas those with basiliximab did not. CMV and fungal infections (*P* < 0.001) and the epoch of transplantation (*P* < 0.001), were associated with TRM at 3 months post-transplantation only.

### Risk factors for early TRM and TRM

The risk factors of early TRM and TRM are shown in Tables 2 and 3, respectively. Based on the Cox multivariate analysis, older age (hazard ratio [HR] = 1.06; *P* < 0.001), CAD (HR = 3.02; *P* = 0.002), cardiac arrhythmia (HR = 1.98; *P* = 0.027), and hemodialysis compared to pre-emptive kidney transplant (HR = 2.53; *P* = 0.046) were independently associated with early TRM. Moreover, older age (HR = 1.07; *P* < 0.001), CAD (HR = 2.88, *P* = < 0.001), and hemodialysis (HR = 2.35, *P* = 0.004) were consistently independent risk factors of TRM at any time. However, fungal infection, (HR = 2.61; *P* < 0.001), and the epoch of transplantation

**Table 1. Comparison of the characteristics between living kidney recipients versus deceased ones at 1 and 3 months after transplantation.**

| Characteristics[a] | Early TRM | | | TRM | | |
|---|---|---|---|---|---|---|
| | Living at 1 month | Death by 1 month | P-value[b] | Living at 3 months | Death by 3 months | P-value[b] |
| Number (%) | 15,999 | 74 | | 15,913 | 160 | |
| Age, years | 47 (37–55) | 56 (48.8–61) | < 0.001 | 47 (37–55) | 55.5 (48–61) | < 0.001 |
| < 50 | 9,188 (57.4) | 19 (25.7) | < 0.001 | 9,163 (57.6) | 44 (27.5) | < 0.001 |
| 50–59 | 4,830 (30.2) | 32 (43.2) | | 4,792 (30.1) | 70 (43.8) | |
| 60–69 | 1,838 (11.5) | 18 (24.3) | | 1,819 (11.4) | 37 (23.1) | |
| 70–79 | 143 (0.9) | 5 (6.8) | | 139 (0.9) | 9 (5.6) | |
| Sex, male | 9,451 (59.1) | 44 (59.5) | 0.946 | 9,403 (59.1) | 92 (57.5) | 0.684 |
| Cause of ESRD | | | | | | |
| Diabetes mellitus | 3,501 (21.9) | 19 (25.7) | 0.431 | 3,479 (21.9) | 41 (25.6) | 0.252 |
| Hypertension | 2,001 (12.5) | 8 (10.8) | 0.660 | 1,992 (12.5) | 17 (10.6) | 0.471 |
| Glomerulonephritis | 2,850 (17.8) | 11 (14.9) | 0.508 | 2,840 (17.8) | 21 (13.1) | 0.120 |
| Cystic kidney disease | 368 (2.3) | 2 (2.7) | 0.818 | 365 (2.3) | 5 (3.1) | 0.485 |
| Underlying disease[c] | | | | | | |
| Cardiac disease | | | | | | |
| Coronary artery disease | 392 (2.5) | 9 (12.2) | < 0.001 | 384 (2.4) | 17 (10.6) | < 0.001 |
| Acute myocardial infarction | 288 (1.8) | 2 (2.7) | 0.561 | 283 (1.8) | 7 (4.4) | 0.014 |
| Cardiac arrhythmia | 1,240 (7.8) | 13 (17.6) | 0.002 | 1,230 (7.7) | 23 (14.4) | 0.002 |
| Cerebrovascular disease | | | | | | |
| Cerebral hemorrhage | 54 (0.3) | 0 (0.0) | 0.617 | 54 (0.3) | 0 (0.0) | 0.460 |
| Cerebral infarction | 247 (1.5) | 1 (1.4) | 0.893 | 246 (1.5) | 2 (1.3) | 0.763 |
| Hemodialysis | 13,134 (82.1) | 69 (93.2) | 0.012 | 13,055 (82.0) | 148 (92.5) | 0.001 |
| Dialysis vintage, months[d] | 42.5 (29.5–62.8) | 16.0 (9.5–24.5) | 0.051 | 41.0 (29.0–63.5) | 24.5 (12.8–39.0) | 0.179 |
| Before steroid use[e] | 1,148 (7.2) | 3 (4.1) | 0.416 | 1,140 (7.2) | 11 (6.9) | 1.000 |
| Induction therapy | | | | | | |
| Basiliximab | 12,637 (79.0) | 55 (74.3) | 0.402 | 12,569 (79.0) | 123 (76.9) | 0.579 |
| Anti-thymocyte globulin | 1,818 (11.4) | 22 (29.7) | < 0.001 | 1,799 (11.3) | 41 (25.6) | < 0.001 |
| Infection | | | | | | |
| CMV infection | 694 (4.3) | 4 (5.4) | 0.653 | 1,900 (11.9) | 37 (23.1) | < 0.001 |
| Fungal infection | 639 (4.0) | 2 (2.7) | 0.571 | 1,205 (7.6) | 37 (23.1) | < 0.001 |
| Epoch of transplantation | | | | | | |
| 2003–2009 | 4,634 (29.0) | 27 (36.5) | 0.155 | 4,594 (28.9) | 67 (41.9) | < 0.001 |
| 2010–2016 | 11,365 (71.0) | 47 (63.5) | | 11,319 (71.1) | 93 (58.1) | |

[a] Data were expressed as number (%) or median (interquartile range).

[b] P value was calculated using chi-square test or Mann–Whitney U test.

[c] In case of the presence of underlying diseases, multiple diseases were designated to one patient.

[d] Data were obtained from the Korean Network for Organ Sharing system.

[e] The use of intravenous steroids such as dexamethasone, and prednisolone within 6 months before transplantation.

Abbreviations: CMV, cytomegalovirus; ESRD, end-stage renal disease; TRM, treatment-related mortality.

(HR = 0.34 for 2010–2016; P < 0.001) were correlated to TRM only. Regarding to the epoch of transplantation, the aged between 50 and 59 years (HR = 0.37, P = 0.005 for early TRM; HR = 0.37, P < 0.001 for TRM), the patients receiving basiliximab as induction therapy (HR = 0.44, P = 0.002 for early TRM; HR = 0.40, P < 0.001 for TRM), and recipients with CMV infection (HR = 0.13, P = 0.040 for early TRM; HR = 0.39, P = 0.005 for TRM) presented better outcome in 2010–2016, when compared to 2003–2009.

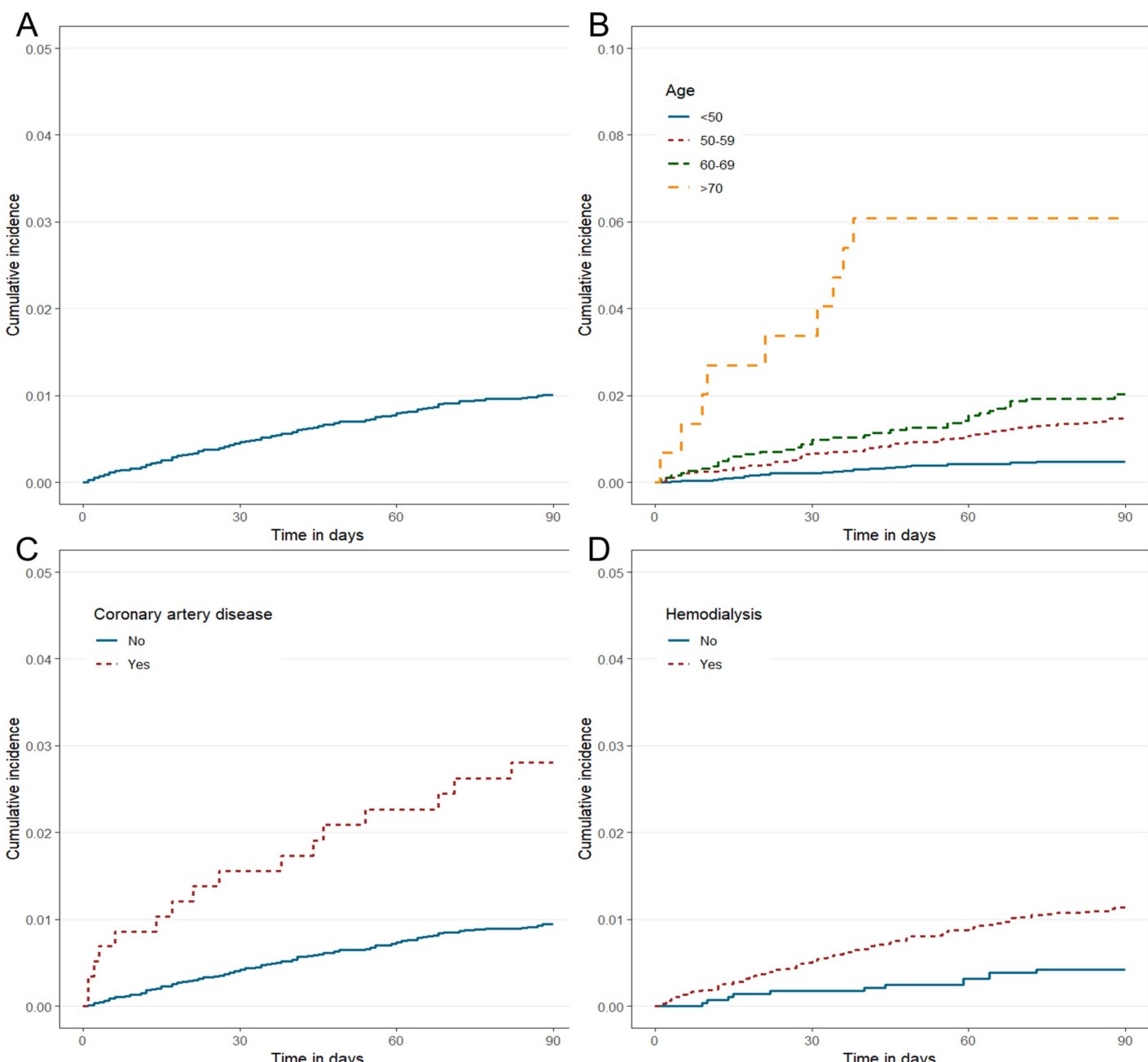

**Fig 1. Cumulative incidence of mortality according to common independent factors of both 1- and 3-month mortality after kidney transplantation.** (A) Total incidence. (B) Older age, (C) Coronary artery disease, and (D) Hemodialysis were associated with worse outcome.

The effect of age on cumulative incidence of mortality is presented in Fig 1B. The older age group presented with higher HRs for both early TRM (50–59 years, 3.21; 60–69 years, 4.74; and 70–79 years, 16.66; $P < 0.001$) and TRM (50–59 years, 3.05; 60–69 years, 4.24; and 70–79 years, 13.16; $P < 0.001$). The effects of CAD and hemodialysis on cumulative incidences are shown in Fig 1C and 1D. In terms of early TRM, a significant difference was observed between patients with a history of cardiac arrhythmia and those without (Fig 2A). Fungal infection (Fig

**Table 2. Univariate and multivariate analyses of 1-month mortality after kidney transplantation.**

| Variable | Univariate | | Multivariate | |
|---|---|---|---|---|
| | HR (95% CI) | *P*-value | HR (95% CI) | *P*-value |
| Age, years[a] | 1.07 (1.05–1.10) | < 0.001 | 1.06 (1.04–1.09) | < 0.001 |
| < 50 | Reference | | | |
| 50–59 | 3.21 (1.82–5.66) | < 0.001 | | |
| 60–69 | 4.74 (2.49–9.03) | < 0.001 | | |
| 70–79 | 16.66 (6.22–44.62) | < 0.001 | | |
| Sex, male | 0.98 (0.62–1.56) | 0.944 | | |
| Cause of ESRD | | | | |
| Diabetes mellitus | 1.22 (0.89–1.75) | 0.451 | | |
| Hypertension | 0.80 (0.58–1.12) | 0.672 | | |
| Glomerulonephritis | 0.93 (0.52–2.15) | 0.591 | | |
| Cystic kidney disease | 1.19 (0.78–2.32) | 0.854 | | |
| Underlying disease | | | | |
| Cardiac disease | | | | |
| Coronary artery disease | 5.51 (2.74–11.06) | < 0.001 | 3.02 (1.48–6.17) | 0.002 |
| Acute myocardial infarction | 1.51 (0.37–6.15) | 0.566 | | |
| Cardiac arrhythmia | 2.53 (1.39–4.60) | 0.002 | 1.98 (1.08–3.62) | 0.027 |
| Cerebrovascular disease | | | | |
| Cerebral hemorrhage | NA | | | |
| Cerebral infarction | 0.87 (0.12–6.26) | 0.890 | | |
| Hemodialysis | 3.00 (1.21–7.45) | 0.018 | 2.53 (1.02–6.28) | 0.046 |
| Dialysis vintage, months[c] | 0.918 (0.833–1.012) | 0.086 | | |
| Before steroid use[d] | 0.55 (0.17–1.74) | 0.307 | | |
| Induction therapy | | | | |
| Basiliximab | 0.77 (0.46–1.30) | 0.326 | | |
| Anti-thymocyte globulin | 3.31 (2.01–5.45) | < 0.001 | 2.62 (1.59–4.32) | < 0.001 |
| Infection | | | | |
| CMV infection | 1.26 (0.46–3.45) | 0.652 | | |
| Fungal infection | 0.66 (0.16–2.71) | 0.569 | | |
| Epoch of transplantation, 2010–2016 | 0.72 (0.45–1.15) | 0.168 | | |

[a] Variables less than 0.05 of *P*-values in univariate analysis were included in the multivariate analysis.

[b] NA is presented if the paucity of deceased or living patients exists for each variable 1 month after kidney transplantation.

[c] Data were obtained from the Korean Network for Organ Sharing system.

[d] The use of intravenous steroids such as dexamethasone, and prednisolone within 6 months before transplantation.

Abbreviations: CI, confidence interval; CMV, cytomegalovirus; ESRD, end-stage renal disease; HR, hazard ratio; NA, not applicable.

2B) affected TRM (after early TRM). The protective effect of transplantation in 2010–2016 is illustrated in Fig 2C.

## Discussion

In the present study, a comprehensive analysis of 1- and 3-month mortality after kidney transplantation in Korea was conducted. Older age, CAD, cardiac arrhythmia, and hemodialysis were risk factors for early TRM. For TRM, older age, CAD, and hemodialysis were common independent risk factors observed in both early TRM and TRM. In contrast, cardiac arrhythmia is a risk factor that associated with early TRM only. Fungal infection and the epoch of transplantation were factors associated with TRM only.

**Table 3. Univariate and multivariate analyses of 3-month mortality after kidney transplantation.**

| Variable | Univariate | | Multivariate | |
|---|---|---|---|---|
| | HR (95% CI) | *P*-value | HR (95% CI) | *P*-value |
| Age, years[a] | 1.07 (1.05–1.09) | < 0.001 | 1.07 (1.05–1.09) | < 0.001 |
| < 50 | | | | |
| 50–59 | 3.05 (2.09–4.44) | < 0.001 | | |
| 60–69 | 4.24 (2.74–6.57) | < 0.001 | | |
| 70–79 | 13.16 (6.43–26.96) | < 0.001 | | |
| Sex, female | 1.07 (0.78–1.46) | 0.690 | | |
| Cause of ESRD | | | | |
| Diabetes mellitus | 1.25 (0.92–1.59) | 0.273 | | |
| Hypertension | 0.86 (0.68–1.10) | 0.463 | | |
| Glomerulonephritis | 0.91 (0.49–1.75) | 0.385 | | |
| Cystic kidney disease | 1.23 (0.87–2.41) | 0.526 | | |
| Underlying disease | | | | |
| Cardiac disease | | | | |
| Coronary artery disease | 4.82 (2.92–7.97) | < 0.001 | 2.88 (1.71–4.84) | < 0.001 |
| Acute myocardial infarction | 2.48 (1.16–5.29) | 0.019 | 1.75 (0.81–3.80) | 0.157 |
| Cardiac arrhythmia | 1.99 (1.28–3.10) | 0.002 | 1.40 (0.89–2.18) | 0.145 |
| Cerebrovascular disease | | | | |
| Cerebral hemorrhage | NA | | | |
| Cerebral infarction | 0.80 (0.20–3.23) | 0.755 | | |
| Hemodialysis | 2.69 (1.49–4.85) | 0.001 | 2.35 (1.30–4.25) | 0.004 |
| Dialysis vintage, months[c] | 0.963 (0.911–1.017) | 0.179 | | |
| Before steroid use[d] | 0.95 (0.52–1.76) | 0.882 | | |
| Induction therapy | | | | |
| Basiliximab | 0.88 (0.61–1.28) | 0.514 | | |
| Anti-thymocyte globulin | 2.73 (1.92–3.90) | < 0.001 | 2.38 (1.62–3.49) | < 0.001 |
| Infection | | | | |
| CMV infection | 2.19 (1.51–3.16) | < 0.001 | 1.39 (0.93–2.08) | 0.106 |
| Fungal infection | 3.57 (2.47–5.15) | < 0.001 | 2.61 (1.79–3.82) | < 0.001 |
| Epoch of transplantation, 2010–2016 | 0.58 (0.42–0.79) | 0.001 | 0.34 (0.24–0.48) | < 0.001 |

[a] Variables less than 0.05 of *P*-values in univariate analysis were included in the multivariate analysis.

[b] NA is presented if the paucity of deceased or living patients exists for each variable 3 months after kidney transplantation.

[c] Data were obtained from the Korean Network for Organ Sharing system.

[d] The use of intravenous steroids such as dexamethasone, and prednisolone within 6 months before transplantation.

Abbreviations: CI, confidence interval; CMV, cytomegalovirus; ESRD, end-stage renal disease; HR, hazard ratio; NA, not applicable.

Cardiovascular disease has been a well-known risk factor and cause of short- and long-term mortality after kidney transplantation [16,24]. Mortality from cardiovascular disease rather than infection has become a more predominant cause of death due to infection control [25]. Atheroma, left ventricular hypertrophy, and vascular calcification were the main mechanisms of cardiovascular disease after kidney transplantation [26]. Regarding CAD, coronary artery calcification was highly prevalent after kidney transplantation [27]. Coronary angiogram is recommended to individuals aged over 50 years who present with DM or previous cardiac events [28]. Cardiac arrhythmia occurred in 30–60% of ESRD patients and was affected by physiologic changes and hemodialysis [29,30]. The use of an implantable cardioverter defibrillator has been recommended if a life-threatening ventricular arrhythmia exists in a patient

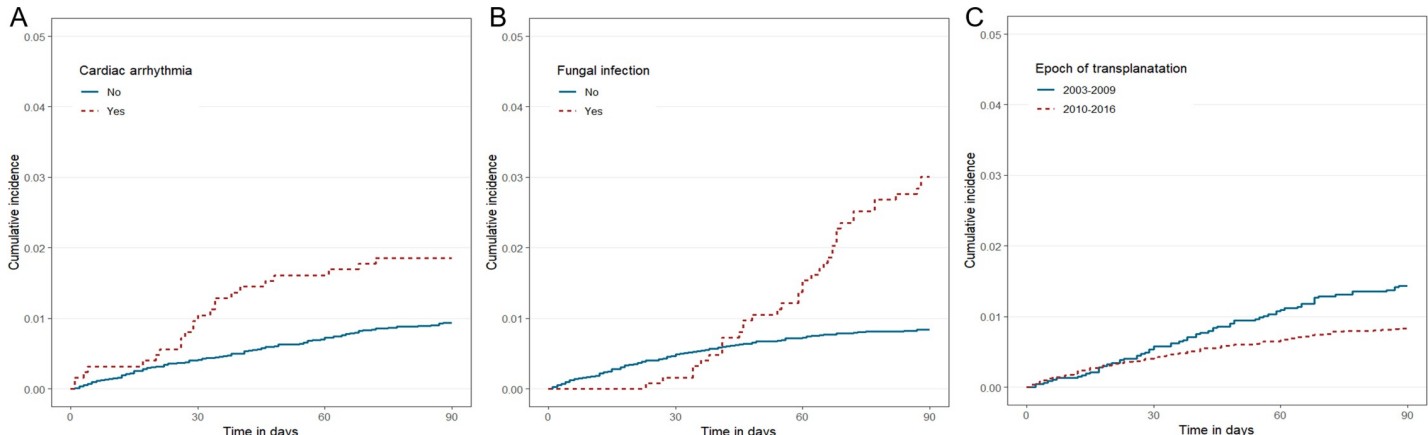

**Fig 2. Cumulative incidence of mortality according to the factors associated 1- or 3-month mortality after kidney transplantation.** (A) Cardiac arrhythmia was related to a worse outcome 1 month after transplantation. (B) Fungal infection were a risk factor of 3-month mortality after transplantation. (C) Recent epoch of transplantation (2010–2016) was a protective factor of 3-month mortality compared to the treatment-related mortality of previous epoch (2003–2009).

who is waiting for kidney transplantation [31]. According to a previous study, graft loss and mortality increased after 1 and 5 years of kidney transplantation in patients with cardiac arrhythmia [32]. Because of these risk factors of TRM, patients who have a history of CAD or cardiac arrhythmia should be counselled for additional work-up and proper management.

Patients without previous hemodialysis showed more favorable outcomes based on our study, despite of discrepancies in preemptive kidney transplantation suggested in previous reports [33,34]. Dialysis-associated comorbidities, decreased immune response, and cardiovascular complications might influence the outcome of non-preemptive kidney transplantation. Prolonged hemodialysis with long waiting times for transplantation has been consistently confirmed to be associated with worse outcomes [35]. The present study revealed that non-preemptive kidney transplantation is related to very short term mortality, such as early TRM and TRM. These findings support early access to transplantation whenever feasible.

The use of anti-thymocyte globulin has been greater in high-risk recipients such as highly sensitized patients, recipients from deceased donors, re-transplantations, and ABO incompatible transplants [36]. According to a prospective, randomized study, patients receiving anti-thymocyte globulin presented a greater incidence of infection (85.8%) compared to those with basiliximab (75.2%) at 12 months after transplantation [37]. However, there was no significant difference in patient survival, similar to the results of a recent study using a network meta-analysis [38]. In Korea, the one-year patient survival in the anti-thymocyte globulin group (89.4%) was compared to the basiliximab group (93.8%), and presented no significant difference [39]. Based on our data, the high-risk recipients receiving anti-thymocyte globulin were significantly associated with early mortality. Further studies for the premature mortality are necessary to validate our results, and intensive care for the high-risk patients receiving anti-thymocyte globulin is important for improving outcomes.

Fungal infections were not common (about 5%) [40] and usually detected after 90 days, however, most infections occurring within 90 days consisted of invasive candidiasis or aspergillosis [41]. Since invasive fungal infections have a mortality rate of 25–30%, these patients require careful management [42]. Obtaining a detailed history of the candidate's risk, as posed by travel and residential exposures, is an important step for prevention and early diagnosis. The risk factors such as triple immunosuppression, broad spectrum antibiotics for more than 2 weeks, and diabetes mellitus should be also noted. Augmented screening, prophylaxis, and

proper work-ups including culture, antigen-based immunoassay, chest radiography, and computed tomography, are all essential to improving the prognosis of kidney recipients [43].

Our risk analysis showed that age was a significant factor ($P < 0.001$) for both early TRM and TRM. The significant association between old age and poor outcome was persistently reported in previous studies [14,18,44], which have to be considered for patient counselling and selection.

Donor status has been a well-known important factor for short- and long-term mortality after kidney transplantation [15,45]. According to previous studies, kidney allograft recipients that died within the first year after transplantation were more likely to be recipients of deceased donor kidneys [18,44]. It was difficult to compare TRM of our cohort with those of other countries directly because of lack of available data. More intensive care for recipients from deceased donors at early point after transplantation is recommended.

The recent year of transplantation was a protective factor for TRM, which is similar to previous studies [4,24,26,46]. This improvement was based on improvements in surgical and anesthesia techniques and methods for immunologic barriers; the development of chemical and biological immunosuppressive drugs, including cyclosporine, mycophenolate mofetil, and tacrolimus [36]; and infection control and appropriate patient selection. The risk of mortality has decreased over the years in most of the categories of patients [26], which is consistent with our results. Even diabetic and old-aged recipients had better outcome. In particular, relatively low- or intermediate-risk patients such as aged 50 to 59 years, and patients receiving basiliximab were influence by the improved protocols, and showed better outcome than high-risk recipients (aged over 60 years, and recipients with anti-thymocyte globulin). Further, more aggressive and sophisticated infection controls on CMV such as monitoring quantitative levels, and high dose of antiviral therapy [47] may protect more recipients in 2010–2016 than those in 2003–2009. However, patients with cardiovascular disease, particularly CAD, should be counselled because their outcome has not improved based on our study and previous reports [26,48].

This study had several limitations. The lack of detailed clinical information, such as donor's characteristics and laboratory data (immunologic antibody profiles, and serology for CMV and fungus), led to restrictions on the analysis of wider variables for TRM. Moreover, classification bias could exist because we used registry data based on physicians' diagnoses. Despite these limitations, the strength of this study includes the use of a nationwide population database of recent kidney recipients. To the best of our knowledge, no other study has reported about TRM and the causes of death using a nationwide data source, particularly in Asia. The relatively large sample size covering the entire national population and unbiased measures used in this study could provide reliable information about kidney recipients.

In conclusion, our study characterized risk factors and causes of 1- and 3-month mortality after kidney transplantation. Old age, particularly greater than 70 years, CAD, and hemodialysis prior to transplant were common risk factors of both early TRM and TRM. By contrast, cardiac arrhythmia was a risk factors for early TRM only, and fungal infection, and epoch of transplantation were important risk factors associated with TRM only. The most common causes of death were chronic kidney disease, cardiovascular disease, and type 2 DM, which require intensive management immediately after transplantation. The risk factors we have identified should be considered when counselling and selecting patients to prevent catastrophic TRM.

## Supporting information

**S1 Table. Data set of recipients with TRM after kidney transplantation.**
(XLSX)

## Acknowledgments

The authors acknowledge the efforts of the staff of the HIRA database, which is supported by the NHI system of Korea, for the maintenance and extraction of data about precise kidney transplantation as a research resource. We also thank Hyun Jung Kim and Hyeong Sik Ahn and the staff of the Department of Preventive Medicine, College of Medicine, Korea University, for their assistance in preparing this article.

## Author Contributions

**Conceptualization:** Ho Sup Lee, Soyoung Ock, Seri Jeong.

**Data curation:** Sangjin Lee, Seri Jeong.

**Formal analysis:** Sangjin Lee, Haeun Oh.

**Funding acquisition:** Seri Jeong.

**Investigation:** Hyunji Choi, Woonhyoung Lee.

**Methodology:** Seom Gim Kong, Da Jung Kim, Taeyun Kim.

**Project administration:** Seri Jeong.

**Resources:** Seom Gim Kong, Da Jung Kim, Seri Jeong.

**Supervision:** Ye Na Kim, Soyoung Ock, Seri Jeong.

**Validation:** Min-Jeong Park, Wonkeun Song.

**Visualization:** Sangjin Lee, John Hoon Rim, Seri Jeong.

**Writing – original draft:** Hyunji Choi, Woonhyoung Lee, Taeyun Kim.

**Writing – review & editing:** John Hoon Rim, Jong-Han Lee, Seri Jeong.

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
