## [Decision Letter · Decision Letter 0]

24 Mar 2020

PONE-D-20-03919

The risk factors associated with treatment-related mortality in 16,073 kidney transplantation - A nationwide cohort study

PLOS ONE

Dear Dr. Seri Jeong,

Thank you for submitting your manuscript to PLOS ONE. After careful consideration, we feel that it has merit but does not fully meet PLOS ONE’s publication criteria as it currently stands. Therefore, we invite you to submit a revised version of the manuscript that addresses the points raised during the review process.

We would appreciate receiving your revised manuscript timely. To enhance the reproducibility of your results, we recommend that if applicable you deposit your laboratory protocols in protocols.io, where a protocol can be assigned its own identifier (DOI) such that it can be cited independently in the future. For instructions see: http://journals.plos.org/plosone/s/submission-guidelines#loc-laboratory-protocols

We look forward to receiving your revised manuscript.

Kind regards,

Academic Editor

PLOS ONE

Additional Editor Comments (if provided):

Please try to address the issues and concerns from the reviewer(s).

Journal Requirements:

Reviewers' comments:

Reviewer's Responses to Questions

**Comments to the Author**

1. Is the manuscript technically sound, and do the data support the conclusions?

Reviewer #1: Partly

2. Has the statistical analysis been performed appropriately and rigorously? 

Reviewer #1: Yes

3. Have the authors made all data underlying the findings in their manuscript fully available?

Reviewer #1: Yes

4. Is the manuscript presented in an intelligible fashion and written in standard English?

Reviewer #1: Yes

5. Review Comments to the Author

Reviewer #1: In this paper submitted by Jeong et al, an investigation about the risk factors causes of TRM after kidney transplantation, focusing on vascular diseases was presented.

The paper is written in a correct and fluent English, and statistical analyses are correctly described and presented by the authors.

Nevertheless, the paper presents several limitations, apart the retrospective design, which make oit unsuitable for the publication in this form. First of all, several important data are missing and in my opinion crucial for the aim of the study: dialysis vintage, prevalence of deceased/living donor (if not considered explain why), basic nephropathy, and steroid therapy before therapy, donor characteristics. All those factors might impact also on the global results of the study that at the moment does not add any novel knowledge on the problem.

In addition many topics need a better clarification and explanation: definition of CMD disease, prevalence of CMV serum-negativity. The cause of death classification is absolutely unreasonable, - “chronic kidney disease was the main cause of both early TRM and TRM, followed cystic kidney disease” ??????

6. PLOS authors have the option to publish the peer review history of their article (what does this mean?). If published, this will include your full peer review and any attached files.

Reviewer #1: No

---

## [Author Response · Author response to Decision Letter 0]

3 May 2020

1. Thank you for providing the following Data Availability Statement:

The repository data for public release is not available because of the personally dentifiable information. The full dataset includes clinic centers in which they attend, insurance conditions. Therefore, concerning privacy risks, the data is managed by authorized executive supervisor. If one researcher asks to access data, the person in charge releases data with blind identification for the discrete requirements and the data should be analyzed only in permitted rooms in centers of National Health Insurance Service. Subsets of data limited to anonymisable information obtained and analyzed during this study are included in this article (tables, figures, and supporting information). Contact information for a data access committee is listed as follows: National Health Insurance Sharing Service, Tel: 82-33-736-2432; Official internet site: https://nhiss.nhis.or.kr/bd/ay/bdaya001iv.do."

Before we proceed, please confirm the following:

1) Please give the full name of the organization that has imposed the data restictions (e.g., a Research Ethics Committee or Institutional Review Board, etc.).

We submitted the security memorandum and pledge to the Institutional Review Board of National Health Insurance Sharing Service when we access these data. The original files of security memorandum and pledge have been uploaded for this revision. The translated contents include “Any data obtained from National Health Insurance Sharing Service will not be taken out externally or used for any other purpose. I pledge to take any civil and criminal penalties.”. Your kind consideration for this security situation would be greatly appreciated. We have inserted additional explanation in to the revised Data Availability Statement (page 20, lines 11 to 13) as follows. 

“If one researcher asks to access data, the researcher should submit the security memorandum and pledge to the Institutional Review Board of National Health Insurance Sharing Service. After approval, the person in charge releases data with blind identification for the discrete requirements and the data should be analyzed only in permitted rooms in centers of National Health Insurance Service.”

2) Please confirm that others would be able to access these data in the same manner as the authors. Please also confirm that the authors did not have any special access privileges that others would not have.

We confirmed that others could access these data in the same manner as the authors and the authors did not have any special access privileges. We also have added these statement to the revised Data Availability Statement (page 20, lines 15 to 16) as follows.

“The other researchers could access these data in the same manner as the authors and the authors did not have any special access privileges.”

3) Please confirm that the data that researchers can access fits our definition of "minimal data set" as outlined here: https://journals.plos.org/plosone/s/data-availability#loc-minimal-data-set-definition

We have provided maximally permitted data for meeting the requirements of “minimal data set”. Although entire data sets of kidney recipients were not permitted, the anonymisable data for recipients with treatment-related mortality focused on this manuscript and used for tables and graphs were provided in S1 Table.

We have checked the PLOS ONE style templates and corrected the location of References section. The revised References section has been listed after the main text, before the supporting information. The file naming also checked and corrected.

We provided the minimal data set, which had anonymisable information in S1 Table. Therefore, we have corrected the sentence from “Subsets of data limited to anonymisable information obtained and analyzed during this study are included in this article (tables, figures, and supporting information) and are available from the corresponding author upon reasonable request.” to “Subsets of data limited to anonymisable information obtained and analyzed during this study are included in this article (tables, figures, and supporting information).” in the revised Data accessibility statement section (page 20, lines 14 to 16). Because National Health Insurance Sharing Service restricts to share full dataset concerning privacy risks, minimal anonymized data set focusing on recipients with treatment-related mortality after kidney transplantation was provided. Contact information for a data access committee and the way for obtaining the data were described in the Data accessibility statement section. We have added these statement to the revised cover letter.

Response to the reviewer’s comments

1. Is the manuscript technically sound, and do the data support the conclusions?

Reviewer #1: Partly

We have corrected and checked that the revised manuscript described a technically sound piece of scientific research and that the data supported the conclusions. We also responded to the reviewer #1’s comments sincerely.

2. Has the statistical analysis been performed appropriately and rigorously? 

Reviewer #1: Yes

We have checked that the statistical analysis has been conducted appropriately and rigorously.

3. Have the authors made all data underlying the findings in their manuscript fully available?

Reviewer #1: Yes

According to the journal requirements, we provided the minimal data set focusing on recipients with treatment-related mortality after kidney transplantation. Contact information for a data access committee and the way for obtaining the data were described in the Data accessibility statement section. We have added these statement to the revised cover letter.

“We collected the following demographic data and baseline characteristics of kidney recipients from the HIRA database: age, sex, medical comorbidities focusing on cardiac and cerebrovascular diseases reported to be important causes of early mortality [16], dialysis status, cytomegalovirus (CMV) and fungal infection, and year of transplantation (S1 Table).”

S1 Table. Data set of recipients with TRM after kidney transplantation.

4. Is the manuscript presented in an intelligible fashion and written in standard English?

Reviewer #1: Yes

We used manuscript editing service before submission.

5. Review Comments to the Author

Response to reviewer #1’s comments

Comment 1: In this paper submitted by Jeong et al, an investigation about the risk factors causes of TRM after kidney transplantation, focusing on vascular diseases was presented.

The paper is written in a correct and fluent English, and statistical analyses are correctly described and presented by the authors.

Nevertheless, the paper presents several limitations, apart the retrospective design, which make oit unsuitable for the publication in this form. First of all, several important data are missing and in my opinion crucial for the aim of the study: dialysis vintage, prevalence of deceased/living donor (if not considered explain why), basic nephropathy, and steroid therapy before therapy, donor characteristics. All those factors might impact also on the global results of the study that at the moment does not add any novel knowledge on the problem.

Response 1: As indicated by reviewer, we have provided available information in the revised manuscript. 

Dialysis vintage: 

We have requested additional data from another database operated by the Korean Network for Organ Sharing system in order to analyze dialysis vintage. In this database, the records of recipients who underwent kidney transplantation in 40 medical centers around the country have been registered. We have provided the results of dialysis vintage, time from dialysis to transplantation, in the revised Tables 1-3 as follows. The source of data was also described in revised Materials and Methods section as follows.

Results

Table 1. Comparison of the characteristics between living kidney recipients versus deceased ones at 1 and 3 months after transplantation.

Characteristicsa Early TRM TRM

 Living at 1 month Death by 1 month P-valueb Living at 3 months Death by 3 months P-valueb

Number (%) 15,999 74 　 15,913 160 　

Age, years 47 (37-55) 56 (48.8-61) < 0.001 47 (37-55) 55.5 (48-61) < 0.001

< 50 9,188 (57.4) 19 (25.7) < 0.001 9,163 (57.6) 44 (27.5) < 0.001

50–59 4,830 (30.2) 32 (43.2) 　 4,792 (30.1) 70 (43.8) 　

60–69 1,838 (11.5) 18 (24.3) 　 1,819 (11.4) 37 (23.1) 　

70–79 143 (0.9) 5 (6.8) 　 139 (0.9) 9 (5.6) 　

Sex, male 9,451 (59.1) 44 (59.5) 0.946 9,403 (59.1) 92 (57.5) 0.684 

Cause of ESRD 

Diabetes mellitus 3,501 (21.9) 19 (25.7) 0.431 3,479 (21.9) 41 (25.6) 0.252

Hypertension 2,001 (12.5) 8 (10.8) 0.660 1,992 (12.5) 17 (10.6) 0.471

Glomerulonephritis 2,850 (17.8) 11 (14.9) 0.508 2,840 (17.8) 21 (13.1) 0.120

Cystic kidney disease 368 (2.3) 2 (2.7) 0.818 365 (2.3) 5 (3.1) 0.485

Underlying diseasec 　 　 　 　 　 　

 Cardiac disease 　 　 　 　 　 　

 Coronary artery disease 392 (2.5) 9 (12.2) < 0.001 384 (2.4) 17 (10.6) < 0.001

 Acute myocardial infarction 288 (1.8) 2 (2.7) 0.561 283 (1.8) 7 (4.4) 0.014 

 Cardiac arrhythmia 1,240 (7.8) 13 (17.6) 0.002 1,230 (7.7) 23 (14.4) 0.002 

 Cerebrovascular disease 　 　 　 　 　 　

 Cerebral hemorrhage 54 (0.3) 0 (0.0) 0.617 54 (0.3) 0 (0.0) 0.460 

 Cerebral infarction 247 (1.5) 1 (1.4) 0.893 246 (1.5) 2 (1.3) 0.763 

Hemodialysis 13,134 (82.1) 69 (93.2) 0.012 13,055 (82.0) 148 (92.5) 0.001 

Dialysis vintage, monthsd 42.5 (29.5-62.8) 16.0 (9.5-24.5) 0.051 41.0 (29.0-63.5) 24.5 (12.8-39.0) 0.179

Before steroid usee 1,148 (7.2) 3 (4.1) 0.416 1,140 (7.2) 11 (6.9) 1.000

Infection 　 　 　 　 　 　

 CMV infection 694 (4.3) 4 (5.4) 0.653 1,900 (11.9) 37 (23.1) < 0.001

 Fungal infection 639 (4.0) 2 (2.7) 0.571 1,205 (7.6) 37 (23.1) < 0.001

Epoch of transplantation 　 　 　 　 　 　

 2003–2009 4,634 (29.0) 27 (36.5) 0.155 4,594 (28.9) 67 (41.9) < 0.001

 2010–2016 11,365 (71.0) 47 (63.5) 　 11,319 (71.1) 93 (58.1) 　

a Data were expressed as number (%) or median (interquartile range).

b P value was calculated using chi-square test or Mann–Whitney U test.

c In case of the presence of underlying diseases, multiple diseases were designated to one patient.

d Data were obtained from the Korean Network for Organ Sharing system.

e The use of intravenous steroids such as dexamethasone, and prednisolone within 6 months before transplantation.

Abbreviations: CMV, cytomegalovirus; ESRD, end-stage renal disease; TRM, treatment-related mortality.

Table 2. Univariate and multivariate analyses of 1-month mortality after kidney transplantation.

Variable Univariate Multivariate

 HR (95% CI) P-value HR (95% CI) P-value

Age, yearsa 1.07 (1.05–1.10) < 0.001 1.07 (1.05-1.10) < 0.001

 < 50 Reference 　 　 　

 50–59 3.21 (1.82-5.66) < 0.001 　 　

 60–69 4.74 (2.49-9.03) < 0.001 　 　

 70–79 16.66 (6.22-44.62) < 0.001 　 　

Sex, male 0.98 (0.62-1.56) 0.944 　 　

Cause of ESRD 

 Diabetes mellitus 1.22 (0.89-1.75) 0.451 

 Hypertension 0.80 (0.58-1.12) 0.672 

 Glomerulonephritis 0.93 (0.52-2.15) 0.591 

 Cystic kidney disease 1.19 (0.78-2.32) 0.854 

Underlying disease 　 　 　 　

 Cardiac disease 　 　 　 　

 Coronary artery disease 5.51 (2.74-11.06) < 0.001 2.81 (1.37-5.78) 0.005 

 Acute myocardial infarction 1.51 (0.37-6.15) 0.566 　 　

 Cardiac arrhythmia 2.53 (1.39-4.60) 0.002 1.99 (1.09-3.64) 0.025 

 Cerebrovascular disease 　 　 　 　

 Cerebral hemorrhage NA 　 　 　

 Cerebral infarction 0.87 (0.12-6.26) 0.890 　 　

Hemodialysis 3.00 (1.21-7.45) 0.018 2.58 (1.04-6.42) 0.041 

Dialysis vintage, monthsc 0.918 (0.833-1.012) 0.086 

Before steroid used 0.55 (0.17-1.74) 0.307 

Infection 　 　 　 　

 CMV infection 1.26 (0.46-3.45) 0.652 　 　

 Fungal infection 0.66 (0.16-2.71) 0.569 　 　

Epoch of transplantation, 2010–2016 0.72 (0.45-1.15) 0.168 　 　

a Variables less than 0.05 of P-values in univariate analysis were included in the multivariate analysis.

b NA is presented if the paucity of deceased or living patients exists for each variable 1 month after kidney transplantation.

c Data were obtained from the Korean Network for Organ Sharing system.

d The use of intravenous steroids such as dexamethasone, and prednisolone within 6 months before transplantation.Abbreviations: CI, confidence interval; CMV, cytomegalovirus; ESRD, end-stage renal disease; HR, hazard ratio; NA, not applicable.

Table 3. Univariate and multivariate analyses of 3-month mortality after kidney transplantation.

Variable Univariate Multivariate

 HR (95% CI) P-value HR (95% CI) P-value

Age, yearsa 1.07 (1.05–1.09) < 0.001 1.08 (1.06-1.09) < 0.001

 < 50 　 　 　 　

 50–59 3.05 (2.09-4.44) < 0.001 　 　

 60–69 4.24 (2.74-6.57) < 0.001 　 　

 70–79 13.16 (6.43-26.96) < 0.001 　 　

Sex, female 1.07 (0.78-1.46) 0.690 　 　

Cause of ESRD 

 Diabetes mellitus 1.25 (0.92-1.59) 0.273 

 Hypertension 0.86 (0.68-1.10) 0.463 

 Glomerulonephritis 0.91 (0.49-1.75) 0.385 

 Cystic kidney disease 1.23 (0.87-2.41) 0.526 

Underlying disease 　 　 　 　

 Cardiac disease 　 　 　 　

 Coronary artery disease 4.82 (2.92-7.97) < 0.001 2.53 (1.49-4.31) 0.001 

 Acute myocardial infarction 2.48 (1.16-5.29) 0.019 1.70 (0.78-3.69) 0.183 

 Cardiac arrhythmia 1.99 (1.28-3.10) 0.002 1.43 (0.91-2.24) 0.117 

 Cerebrovascular disease 　 　 　 　

 Cerebral hemorrhage NA 　 　 　

 Cerebral infarction 0.80 (0.20-3.23) 0.755 　 　

Hemodialysis 2.69 (1.49-4.85) 0.001 2.32 (1.28-4.19) 0.005 

Dialysis vintage, monthsc 0.963 (0.911-1.017) 0.179 

Before steroid used 0.95 (0.52-1.76) 0.882 

Infection 　 　 　 　

 CMV infection 2.19 (1.51-3.16) < 0.001 1.65 (1.12-2.42) 0.012 

 Fungal infection 3.57 (2.47-5.15) < 0.001 2.48 (1.69-3.65) < 0.001

Epoch of transplantation, 2010–2016 0.58 (0.42-0.79) 0.001 0.43 (0.31-0.60) < 0.001

a Variables less than 0.05 of P-values in univariate analysis were included in the multivariate analysis.

b NA is presented if the paucity of deceased or living patients exists for each variable 3 months after kidney transplantation.

c Data were obtained from the Korean Network for Organ Sharing system.

d The use of intravenous steroids such as dexamethasone, and prednisolone within 6 months before transplantation.

Abbreviations: CI, confidence interval; CMV, cytomegalovirus; ESRD, end-stage renal disease; HR, hazard ratio; NA, not applicable.

Materials and Methods section (page 7, lines 11 to 14)

The data for dialysis vintage, and donor state omitting in HIRA database were obtained from another database operated by the Korean Network for Organ Sharing system. In this database, the records of recipients who underwent kidney transplantation in 40 medical centers around the country were registered.

Prevalence of deceased/living donor (if not considered explain why):

We have also requested data of donor state from the Korean Network for Organ Sharing system since donor information was not included in the Health Insurance Review and Assessment Service (HIRA). The results of living, deceased, and non-heart beating donors are presented in the revised Results section as follows. These variables did not insert into the Tables because the direct combination of data was not available for the multivariate analyses.

Results

Characteristics of patients (page 8, lines 11 to 13)

“Most patients received kidney from living donor (62.2%), followed by deceased (37.5%) and non-heart beating (0.3%) donors.”

Treatment-related mortality (page 11, lines 9 to 12)

“The rates of recipients who died 1 month (1.2% for living, 2.4% for deceased, and 3.3% for non-heart beating) and 3 months (3.2% for living, 5.5% for deceased, and 3.3% for non-heart beating) after transplantation showed significant difference according to the donor state (P < 0.001).”

Discussion (page 17, lines 22 to page 18 lines 3)

“Donor status has been a well-known important factor for short- and long-term mortality after kidney transplantation [15,21]. According to previous studies, kidney allograft recipients that died within the first year after transplantation were more likely to be recipients of deceased donor kidneys [18,20]. It was difficult to compare TRM of our cohort with those of other countries directly because of lack of available data. More intensive care for recipients from deceased donors at early point after transplantation is recommended.”

Basic nephropathy: 

We have added the results of basic nephropathy including diabetes mellitus, hypertension, glomerulonephritis, and cystic kidney disease to the revised Tables 1-3 as follows. 

Table 1. Comparison of the characteristics between living kidney recipients versus deceased ones at 1 and 3 months after transplantation.

Characteristicsa Early TRM TRM

 Living at 1 month Death by 1 month P-valueb Living at 3 months Death by 3 months P-valueb

Number (%) 15,999 74 　 15,913 160 　

Age, years 47 (37-55) 56 (48.8-61) < 0.001 47 (37-55) 55.5 (48-61) < 0.001

< 50 9,188 (57.4) 19 (25.7) < 0.001 9,163 (57.6) 44 (27.5) < 0.001

50–59 4,830 (30.2) 32 (43.2) 　 4,792 (30.1) 70 (43.8) 　

60–69 1,838 (11.5) 18 (24.3) 　 1,819 (11.4) 37 (23.1) 　

70–79 143 (0.9) 5 (6.8) 　 139 (0.9) 9 (5.6) 　

Sex, male 9,451 (59.1) 44 (59.5) 0.946 9,403 (59.1) 92 (57.5) 0.684 

Cause of ESRD 

Diabetes mellitus 3,501 (21.9) 19 (25.7) 0.431 3,479 (21.9) 41 (25.6) 0.252

Hypertension 2,001 (12.5) 8 (10.8) 0.660 1,992 (12.5) 17 (10.6) 0.471

Glomerulonephritis 2,850 (17.8) 11 (14.9) 0.508 2,840 (17.8) 21 (13.1) 0.120

Cystic kidney disease 368 (2.3) 2 (2.7) 0.818 365 (2.3) 5 (3.1) 0.485

Underlying diseasec 　 　 　 　 　 　

 Cardiac disease 　 　 　 　 　 　

 Coronary artery disease 392 (2.5) 9 (12.2) < 0.001 384 (2.4) 17 (10.6) < 0.001

 Acute myocardial infarction 288 (1.8) 2 (2.7) 0.561 283 (1.8) 7 (4.4) 0.014 

 Cardiac arrhythmia 1,240 (7.8) 13 (17.6) 0.002 1,230 (7.7) 23 (14.4) 0.002 

 Cerebrovascular disease 　 　 　 　 　 　

 Cerebral hemorrhage 54 (0.3) 0 (0.0) 0.617 54 (0.3) 0 (0.0) 0.460 

 Cerebral infarction 247 (1.5) 1 (1.4) 0.893 246 (1.5) 2 (1.3) 0.763 

Hemodialysis 13,134 (82.1) 69 (93.2) 0.012 13,055 (82.0) 148 (92.5) 0.001 

Dialysis vintage, monthsd 42.5 (29.5-62.8) 16.0 (9.5-24.5) 0.051 41.0 (29.0-63.5) 24.5 (12.8-39.0) 0.179

Before steroid usee 1,148 (7.2) 3 (4.1) 0.416 1,140 (7.2) 11 (6.9) 1.000

Infection 　 　 　 　 　 　

 CMV infection 694 (4.3) 4 (5.4) 0.653 1,900 (11.9) 37 (23.1) < 0.001

 Fungal infection 639 (4.0) 2 (2.7) 0.571 1,205 (7.6) 37 (23.1) < 0.001

Epoch of transplantation 　 　 　 　 　 　

 2003–2009 4,634 (29.0) 27 (36.5) 0.155 4,594 (28.9) 67 (41.9) < 0.001

 2010–2016 11,365 (71.0) 47 (63.5) 　 11,319 (71.1) 93 (58.1) 　

a Data were expressed as number (%) or median (interquartile range).

b P value was calculated using chi-square test or Mann–Whitney U test.

c In case of the presence of underlying diseases, multiple diseases were designated to one patient.

d Data were obtained from the Korean Network for Organ Sharing system.

e The use of intravenous steroids such as dexamethasone, and prednisolone within 6 months before transplantation.

Abbreviations: CMV, cytomegalovirus; ESRD, end-stage renal disease; TRM, treatment-related mortality.

Table 2. Univariate and multivariate analyses of 1-month mortality after kidney transplantation.

Variable Univariate Multivariate

 HR (95% CI) P-value HR (95% CI) P-value

Age, yearsa 1.07 (1.05–1.10) < 0.001 1.07 (1.05-1.10) < 0.001

 < 50 Reference 　 　 　

 50–59 3.21 (1.82-5.66) < 0.001 　 　

 60–69 4.74 (2.49-9.03) < 0.001 　 　

 70–79 16.66 (6.22-44.62) < 0.001 　 　

Sex, male 0.98 (0.62-1.56) 0.944 　 　

Cause of ESRD 

 Diabetes mellitus 1.22 (0.89-1.75) 0.451 

 Hypertension 0.80 (0.58-1.12) 0.672 

 Glomerulonephritis 0.93 (0.52-2.15) 0.591 

 Cystic kidney disease 1.19 (0.78-2.32) 0.854 

Underlying disease 　 　 　 　

 Cardiac disease 　 　 　 　

 Coronary artery disease 5.51 (2.74-11.06) < 0.001 2.81 (1.37-5.78) 0.005 

 Acute myocardial infarction 1.51 (0.37-6.15) 0.566 　 　

 Cardiac arrhythmia 2.53 (1.39-4.60) 0.002 1.99 (1.09-3.64) 0.025 

 Cerebrovascular disease 　 　 　 　

 Cerebral hemorrhage NA 　 　 　

 Cerebral infarction 0.87 (0.12-6.26) 0.890 　 　

Hemodialysis 3.00 (1.21-7.45) 0.018 2.58 (1.04-6.42) 0.041 

Dialysis vintage, monthsc 0.918 (0.833-1.012) 0.086 

Before steroid used 0.55 (0.17-1.74) 0.307 

Infection 　 　 　 　

 CMV infection 1.26 (0.46-3.45) 0.652 　 　

 Fungal infection 0.66 (0.16-2.71) 0.569 　 　

Epoch of transplantation, 2010–2016 0.72 (0.45-1.15) 0.168 　 　

a Variables less than 0.05 of P-values in univariate analysis were included in the multivariate analysis.

b NA is presented if the paucity of deceased or living patients exists for each variable 1 month after kidney transplantation.

c Data were obtained from the Korean Network for Organ Sharing system.

d The use of intravenous steroids such as dexamethasone, and prednisolone within 6 months before transplantation.Abbreviations: CI, confidence interval; CMV, cytomegalovirus; ESRD, end-stage renal disease; HR, hazard ratio; NA, not applicable.

Table 3. Univariate and multivariate analyses of 3-month mortality after kidney transplantation.

Variable Univariate Multivariate

 HR (95% CI) P-value HR (95% CI) P-value

Age, yearsa 1.07 (1.05–1.09) < 0.001 1.08 (1.06-1.09) < 0.001

 < 50 　 　 　 　

 50–59 3.05 (2.09-4.44) < 0.001 　 　

 60–69 4.24 (2.74-6.57) < 0.001 　 　

 70–79 13.16 (6.43-26.96) < 0.001 　 　

Sex, female 1.07 (0.78-1.46) 0.690 　 　

Cause of ESRD 

 Diabetes mellitus 1.25 (0.92-1.59) 0.273 

 Hypertension 0.86 (0.68-1.10) 0.463 

 Glomerulonephritis 0.91 (0.49-1.75) 0.385 

 Cystic kidney disease 1.23 (0.87-2.41) 0.526 

Underlying disease 　 　 　 　

 Cardiac disease 　 　 　 　

 Coronary artery disease 4.82 (2.92-7.97) < 0.001 2.53 (1.49-4.31) 0.001 

 Acute myocardial infarction 2.48 (1.16-5.29) 0.019 1.70 (0.78-3.69) 0.183 

 Cardiac arrhythmia 1.99 (1.28-3.10) 0.002 1.43 (0.91-2.24) 0.117 

 Cerebrovascular disease 　 　 　 　

 Cerebral hemorrhage NA 　 　 　

 Cerebral infarction 0.80 (0.20-3.23) 0.755 　 　

Hemodialysis 2.69 (1.49-4.85) 0.001 2.32 (1.28-4.19) 0.005 

Dialysis vintage, monthsc 0.963 (0.911-1.017) 0.179 

Before steroid used 0.95 (0.52-1.76) 0.882 

Infection 　 　 　 　

 CMV infection 2.19 (1.51-3.16) < 0.001 1.65 (1.12-2.42) 0.012 

 Fungal infection 3.57 (2.47-5.15) < 0.001 2.48 (1.69-3.65) < 0.001

Epoch of transplantation, 2010–2016 0.58 (0.42-0.79) 0.001 0.43 (0.31-0.60) < 0.001

a Variables less than 0.05 of P-values in univariate analysis were included in the multivariate analysis.

b NA is presented if the paucity of deceased or living patients exists for each variable 3 months after kidney transplantation.

c Data were obtained from the Korean Network for Organ Sharing system.

d The use of intravenous steroids such as dexamethasone, and prednisolone within 6 months before transplantation.

Abbreviations: CI, confidence interval; CMV, cytomegalovirus; ESRD, end-stage renal disease; HR, hazard ratio; NA, not applicable.

Steroid therapy before therapy:

We have inserted the results of steroid therapy before transplantation into the revised Tables 1-3 as follows. The recipients with the use of intravenous steroids such as dexamethasone, and prednisolone within 6 months before transplantation were designated and described in the revised footnotes of Tables.

Table 1. Comparison of the characteristics between living kidney recipients versus deceased ones at 1 and 3 months after transplantation.

Characteristicsa Early TRM TRM

 Living at 1 month Death by 1 month P-valueb Living at 3 months Death by 3 months P-valueb

Number (%) 15,999 74 　 15,913 160 　

Age, years 47 (37-55) 56 (48.8-61) < 0.001 47 (37-55) 55.5 (48-61) < 0.001

< 50 9,188 (57.4) 19 (25.7) < 0.001 9,163 (57.6) 44 (27.5) < 0.001

50–59 4,830 (30.2) 32 (43.2) 　 4,792 (30.1) 70 (43.8) 　

60–69 1,838 (11.5) 18 (24.3) 　 1,819 (11.4) 37 (23.1) 　

70–79 143 (0.9) 5 (6.8) 　 139 (0.9) 9 (5.6) 　

Sex, male 9,451 (59.1) 44 (59.5) 0.946 9,403 (59.1) 92 (57.5) 0.684 

Cause of ESRD 

Diabetes mellitus 3,501 (21.9) 19 (25.7) 0.431 3,479 (21.9) 41 (25.6) 0.252

Hypertension 2,001 (12.5) 8 (10.8) 0.660 1,992 (12.5) 17 (10.6) 0.471

Glomerulonephritis 2,850 (17.8) 11 (14.9) 0.508 2,840 (17.8) 21 (13.1) 0.120

Cystic kidney disease 368 (2.3) 2 (2.7) 0.818 365 (2.3) 5 (3.1) 0.485

Underlying diseasec 　 　 　 　 　 　

 Cardiac disease 　 　 　 　 　 　

 Coronary artery disease 392 (2.5) 9 (12.2) < 0.001 384 (2.4) 17 (10.6) < 0.001

 Acute myocardial infarction 288 (1.8) 2 (2.7) 0.561 283 (1.8) 7 (4.4) 0.014 

 Cardiac arrhythmia 1,240 (7.8) 13 (17.6) 0.002 1,230 (7.7) 23 (14.4) 0.002 

 Cerebrovascular disease 　 　 　 　 　 　

 Cerebral hemorrhage 54 (0.3) 0 (0.0) 0.617 54 (0.3) 0 (0.0) 0.460 

 Cerebral infarction 247 (1.5) 1 (1.4) 0.893 246 (1.5) 2 (1.3) 0.763 

Hemodialysis 13,134 (82.1) 69 (93.2) 0.012 13,055 (82.0) 148 (92.5) 0.001 

Dialysis vintage, monthsd 42.5 (29.5-62.8) 16.0 (9.5-24.5) 0.051 41.0 (29.0-63.5) 24.5 (12.8-39.0) 0.179

Before steroid usee 1,148 (7.2) 3 (4.1) 0.416 1,140 (7.2) 11 (6.9) 1.000

Infection 　 　 　 　 　 　

 CMV infection 694 (4.3) 4 (5.4) 0.653 1,900 (11.9) 37 (23.1) < 0.001

 Fungal infection 639 (4.0) 2 (2.7) 0.571 1,205 (7.6) 37 (23.1) < 0.001

Epoch of transplantation 　 　 　 　 　 　

 2003–2009 4,634 (29.0) 27 (36.5) 0.155 4,594 (28.9) 67 (41.9) < 0.001

 2010–2016 11,365 (71.0) 47 (63.5) 　 11,319 (71.1) 93 (58.1) 　

a Data were expressed as number (%) or median (interquartile range).

b P value was calculated using chi-square test or Mann–Whitney U test.

c In case of the presence of underlying diseases, multiple diseases were designated to one patient.

d Data were obtained from the Korean Network for Organ Sharing system.

e The use of intravenous steroids such as dexamethasone, and prednisolone within 6 months before transplantation.

Abbreviations: CMV, cytomegalovirus; ESRD, end-stage renal disease; TRM, treatment-related mortality.

Table 2. Univariate and multivariate analyses of 1-month mortality after kidney transplantation.

Variable Univariate Multivariate

 HR (95% CI) P-value HR (95% CI) P-value

Age, yearsa 1.07 (1.05–1.10) < 0.001 1.07 (1.05-1.10) < 0.001

 < 50 Reference 　 　 　

 50–59 3.21 (1.82-5.66) < 0.001 　 　

 60–69 4.74 (2.49-9.03) < 0.001 　 　

 70–79 16.66 (6.22-44.62) < 0.001 　 　

Sex, male 0.98 (0.62-1.56) 0.944 　 　

Cause of ESRD 

 Diabetes mellitus 1.22 (0.89-1.75) 0.451 

 Hypertension 0.80 (0.58-1.12) 0.672 

 Glomerulonephritis 0.93 (0.52-2.15) 0.591 

 Cystic kidney disease 1.19 (0.78-2.32) 0.854 

Underlying disease 　 　 　 　

 Cardiac disease 　 　 　 　

 Coronary artery disease 5.51 (2.74-11.06) < 0.001 2.81 (1.37-5.78) 0.005 

 Acute myocardial infarction 1.51 (0.37-6.15) 0.566 　 　

 Cardiac arrhythmia 2.53 (1.39-4.60) 0.002 1.99 (1.09-3.64) 0.025 

 Cerebrovascular disease 　 　 　 　

 Cerebral hemorrhage NA 　 　 　

 Cerebral infarction 0.87 (0.12-6.26) 0.890 　 　

Hemodialysis 3.00 (1.21-7.45) 0.018 2.58 (1.04-6.42) 0.041 

Dialysis vintage, monthsc 0.918 (0.833-1.012) 0.086 

Before steroid used 0.55 (0.17-1.74) 0.307 

Infection 　 　 　 　

 CMV infection 1.26 (0.46-3.45) 0.652 　 　

 Fungal infection 0.66 (0.16-2.71) 0.569 　 　

Epoch of transplantation, 2010–2016 0.72 (0.45-1.15) 0.168 　 　

a Variables less than 0.05 of P-values in univariate analysis were included in the multivariate analysis.

b NA is presented if the paucity of deceased or living patients exists for each variable 1 month after kidney transplantation.

c Data were obtained from the Korean Network for Organ Sharing system.

d The use of intravenous steroids such as dexamethasone, and prednisolone within 6 months before transplantation.Abbreviations: CI, confidence interval; CMV, cytomegalovirus; ESRD, end-stage renal disease; HR, hazard ratio; NA, not applicable.

Table 3. Univariate and multivariate analyses of 3-month mortality after kidney transplantation.

Variable Univariate Multivariate

 HR (95% CI) P-value HR (95% CI) P-value

Age, yearsa 1.07 (1.05–1.09) < 0.001 1.08 (1.06-1.09) < 0.001

 < 50 　 　 　 　

 50–59 3.05 (2.09-4.44) < 0.001 　 　

 60–69 4.24 (2.74-6.57) < 0.001 　 　

 70–79 13.16 (6.43-26.96) < 0.001 　 　

Sex, female 1.07 (0.78-1.46) 0.690 　 　

Cause of ESRD 

 Diabetes mellitus 1.25 (0.92-1.59) 0.273 

 Hypertension 0.86 (0.68-1.10) 0.463 

 Glomerulonephritis 0.91 (0.49-1.75) 0.385 

 Cystic kidney disease 1.23 (0.87-2.41) 0.526 

Underlying disease 　 　 　 　

 Cardiac disease 　 　 　 　

 Coronary artery disease 4.82 (2.92-7.97) < 0.001 2.53 (1.49-4.31) 0.001 

 Acute myocardial infarction 2.48 (1.16-5.29) 0.019 1.70 (0.78-3.69) 0.183 

 Cardiac arrhythmia 1.99 (1.28-3.10) 0.002 1.43 (0.91-2.24) 0.117 

 Cerebrovascular disease 　 　 　 　

 Cerebral hemorrhage NA 　 　 　

 Cerebral infarction 0.80 (0.20-3.23) 0.755 　 　

Hemodialysis 2.69 (1.49-4.85) 0.001 2.32 (1.28-4.19) 0.005 

Dialysis vintage, monthsc 0.963 (0.911-1.017) 0.179 

Before steroid used 0.95 (0.52-1.76) 0.882 

Infection 　 　 　 　

 CMV infection 2.19 (1.51-3.16) < 0.001 1.65 (1.12-2.42) 0.012 

 Fungal infection 3.57 (2.47-5.15) < 0.001 2.48 (1.69-3.65) < 0.001

Epoch of transplantation, 2010–2016 0.58 (0.42-0.79) 0.001 0.43 (0.31-0.60) < 0.001

a Variables less than 0.05 of P-values in univariate analysis were included in the multivariate analysis.

b NA is presented if the paucity of deceased or living patients exists for each variable 3 months after kidney transplantation.

c Data were obtained from the Korean Network for Organ Sharing system.

d The use of intravenous steroids such as dexamethasone, and prednisolone within 6 months before transplantation.

Abbreviations: CI, confidence interval; CMV, cytomegalovirus; ESRD, end-stage renal disease; HR, hazard ratio; NA, not applicable.

Donor characteristics:

Unfortunately, the donor characteristics were not provided by National Health Insurance Sharing Service. We have described this limitation in the revised Discussion section (page 18, lines 14 to 15) as follows. 

“This study had several limitations. The lack of detailed clinical information, such as donor’s characteristics and laboratory data (immunologic antibody profiles, and serology for CMV and fungus), led to restrictions on the analysis of wider variables for TRM.”

Comment 2: In addition many topics need a better clarification and explanation: definition of CMD disease, prevalence of CMV serum-negativity. The cause of death classification is absolutely unreasonable, - “chronic kidney disease was the main cause of both early TRM and TRM, followed cystic kidney disease” ??????

Response 2: We have added explanation for the definition of CMV disease to the revised Materials Methods section (page 6, lines 16 to 17) as follows. 

“The ICD-10-CM codes for CMV disease were B27.1, B25.0, B25.1, B25.8, and B25.9.”

The prevalence of CMV serum-negativity could not be provided because laboratory values for CMV infection were not included in the datasets provided by National Health Insurance Sharing Service. We have described this limitation in the revised Discussion section (page 18, lines 14 to 15) as follows. 

“This study had several limitations. The lack of detailed clinical information, such as donor’s characteristics and laboratory data (immunologic antibody profiles, and serology for CMV and fungus), led to restrictions on the analysis of wider variables for TRM.”

We have eliminated the contents for the cause of death throughout the revised manuscript (Abstract, Materials and Methods, Results [Table 4, and S2 Table], and Discussion sections).

We have uploaded our figure files (Fig 1.tif, and Fig 2.tif) to the PACE digital diagnostic tool to meet PLOS requirements. The preview files (Preview_20200420051917202.pdf, and Preview_20200420052011519.pdf) were generated and checked.

---

## [Decision Letter · Decision Letter 1]

25 May 2020

PONE-D-20-03919R1

The risk factors associated with treatment-related mortality in 16,073 kidney transplantation - A nationwide cohort study

PLOS ONE

Dear Dr. Jeong,

Thank you for submitting your manuscript to PLOS ONE. After careful consideration, we feel that it has merit but does not fully meet PLOS ONE’s publication criteria as it currently stands. Therefore, we invite you to submit a revised version of the manuscript that addresses the points raised during the review process.

Please address the issues by the reviewers to make the next revision.

We look forward to receiving your revised manuscript.

Kind regards,

Academic Editor

PLOS ONE

Reviewers' comments:

Reviewer's Responses to Questions

**Comments to the Author**

1. If the authors have adequately addressed your comments raised in a previous round of review and you feel that this manuscript is now acceptable for publication, you may indicate that here to bypass the “Comments to the Author” section, enter your conflict of interest statement in the “Confidential to Editor” section, and submit your "Accept" recommendation.

Reviewer #1: All comments have been addressed

Reviewer #2: All comments have been addressed

2. Is the manuscript technically sound, and do the data support the conclusions?

Reviewer #1: (No Response)

Reviewer #2: Yes

3. Has the statistical analysis been performed appropriately and rigorously? 

Reviewer #1: (No Response)

Reviewer #2: Yes

4. Have the authors made all data underlying the findings in their manuscript fully available?

Reviewer #1: (No Response)

Reviewer #2: No

5. Is the manuscript presented in an intelligible fashion and written in standard English?

Reviewer #1: (No Response)

Reviewer #2: Yes

6. Review Comments to the Author

Reviewer #1: I thank the authors to have addressed all my recommendations. At the moment the paper is suitable for publication.

Reviewer #2: Treatment-related mortality (TRM) after renal transplantation is a concept different from disease-related mortality and appeared to be a very prevalent entity.

This article would be a valuable contribution to the medical literature to encourage further discussion on this entity.

The writing is clear and easily understandable.

The Authors have worked hard to improve this article, and they have met all criticisms raised by referees

Strengths:

- There are scarce data in scientific literature about TRM within 1 or 3 months after kidney transplantation.

- Authors collected an important amount of data from a very large cohort of patients using a national population based database, which included information about a total of 16,073 kidney recipients.

Specific comments:

- Abstract. Authors should clearly explain the following sentence: “Based on a multivariate analysis, older age (hazard ratio [HR] = 1.07; P < 0.001), coronary artery disease (HR = 2.81; P = 0.005), and hemodialysis (HR = 2.58; P = 0.041) were the risk factors for early TRM.” What do you mean with hemodialysis as a risk factor for early TRM? Patients who underwent to hemodialysis immediately after renal transplant for a DGF, o hemodialysis compared with peritoneal dialysis or pre-emptive renal transplant? This is not clear.

- Introduction, page 4, line 16. Supporting references at the end of the following sentence are needed: “however, studies about 1- or 3-month mortality were extremely limited.”

- Introduction, page 5, line 2. Authors should clearly explain the following sentence: “… to investigate the risk factors and causes of TRM after kidney transplantation focusing on vascular diseases.” Are Authors focused on vascular diseases in this analysis?

- Methods. Authors collected data of the post-transplant administration of antiviral agent, but there is no mention to induction (basiliximab vs thymoglobuline) immunosuppressive therapy, that is supposed to strongly impact on treatment related mortality. This is an important point that Authors should add into the analysis. Otherwise this will represent an important limitation.

- Results. A sub-analysis might be performed distinguishing TRM analysis between transplantation cases performed in different epoch (2003-2009 vs 2010-2016). This might be interesting even if this factor did not reach statistical significance at univariate analysis. In fact, “Epoch of transplantation 2010–2016” showed a trend towards statistical significance.

- Results. In the following sentence, Authors should specify the number of recipients who died among the total number of recipients for each type of transplantation, and only after the percentage value in parentheses: “The rates of recipients who died 1 month (1.2% for living, 2.4% for deceased, and

3.3% for non-heart beating) and 3 months (3.2% for living, 5.5% for deceased, and

3.3% for non-heart beating) after transplantation showed significant difference

according to the donor state (P < 0.001).”

7. PLOS authors have the option to publish the peer review history of their article (what does this mean?). If published, this will include your full peer review and any attached files.

Reviewer #1: No

Reviewer #2: No

---

## [Author Response · Author response to Decision Letter 1]

23 Jun 2020

Response to the reviewer’s comments

Response to reviewer #1’s comments

1. I thank the authors to have addressed all my recommendations. At the moment the paper is suitable for publication.

We thank the reviewer for the constructive review of our manuscript. 

Response to reviewer #2’s comments

Treatment-related mortality (TRM) after renal transplantation is a concept different from disease-related mortality and appeared to be a very prevalent entity.

This article would be a valuable contribution to the medical literature to encourage further discussion on this entity.

The writing is clear and easily understandable.

The Authors have worked hard to improve this article, and they have met all criticisms raised by referees

Strengths:

- There are scarce data in scientific literature about TRM within 1 or 3 months after kidney transplantation.

- Authors collected an important amount of data from a very large cohort of patients using a national population based database, which included information about a total of 16,073 kidney recipients.

Specific comments:

1. Abstract. Authors should clearly explain the following sentence: “Based on a multivariate analysis, older age (hazard ratio [HR] = 1.07; P < 0.001), coronary artery disease (HR = 2.81; P = 0.005), and hemodialysis (HR = 2.58; P = 0.041) were the risk factors for early TRM.” What do you mean with hemodialysis as a risk factor for early TRM? Patients who underwent to hemodialysis immediately after renal transplant for a DGF, o hemodialysis compared with peritoneal dialysis or pre-emptive renal transplant? This is not clear.

We have provided detailed description of hemodialysis in the revised Abstract section as follows for clarification. In addition, we have also inserted this description into the revised Results section as follows.

Abstract section (page 3, lines 8 to 11)

“Based on a multivariate analysis, older age (hazard ratio [HR] = 1.06; P < 0.001), coronary artery disease (HR = 3.02; P = 0.002), and hemodialysis compared with pre-emptive kidney transplantation (HR = 2.53; P = 0.046) were the risk factors for early TRM.”

Results section (page 11, lines 22 to page 12, lines 2)

“Based on the Cox multivariate analysis, older age (hazard ratio [HR] = 1.06; P < 0.001), CAD (HR = 3.02; P = 0.002), cardiac arrhythmia (HR = 1.98; P = 0.027), and hemodialysis compared to pre-emptive kidney transplant (HR = 2.53; P = 0.046) were independently associated with early TRM.”

2. Introduction, page 4, line 16. Supporting references at the end of the following sentence are needed: “however, studies about 1- or 3-month mortality were extremely limited.”

To the best of our knowledge, mortality after kidney transplantation within 1- or 3- months has been seldom reported. There was one report comparing mortality risk between deceased-donor kidney allograft recipients and wait-listed transplant candidates from time of listing published in 1993. Relative risk of mortality within the first 30 days, days 31-365 and more than 365-days post transplantation were 2.43, 0.96 and 0.36, respectively. Another report recently published was dealt with 30-day mortality and compared mostly the ethnic difference of England, and New York State. We have cited these two articles at the end of the sentence in the revised Introduction section as follows. These articles have been presented in the revised Reference section as follows.

Introducton section (page 4, lines 15 to 17)

“Most reports have shown the results of kidney transplantation after 1 [18], 5 [16], and greater than 10 years [19]; however, studies about 1- or 3-month mortality were extremely limited [20,21].”

Reference section (page 23, lines 20 to page 24, lines 1)

“20. Port FK, Wolfe RA, Mauger EA, Berling DP, Jiang K. Comparison of survival probabilities for dialysis patients vs cadaveric renal transplant recipients. JAMA. 1993;270: 1339-1343.

21. Tahir S, Gillott H, Jackson-Spence F, Nath J, Mytton J, Evison F, et al. Do outcomes after kidney transplantation differ for black patients in England versus New York State? A comparative, population-cohort analysis. BMJ Open. 2017;7: e014069.”

3. Introduction, page 5, line 2. Authors should clearly explain the following sentence: “… to investigate the risk factors and causes of TRM after kidney transplantation focusing on vascular diseases.” Are Authors focused on vascular diseases in this analysis?

As indicated by the reviewer, various risk factors related to treatment-related mortality (TRM) were identified in our manuscript. Therefore, we have eliminated “focusing on vascular diseases” in the revised Introduction section (page 5, lines 1 to 2) as follows.

“Using this database, we performed a comprehensive population-based analysis to investigate the risk factors and causes of TRM after kidney transplantation.”

4. Methods. Authors collected data of the post-transplant administration of antiviral agent, but there is no mention to induction (basiliximab vs thymoglobuline) immunosuppressive therapy, that is supposed to strongly impact on treatment related mortality. This is an important point that Authors should add into the analysis. Otherwise this will represent an important limitation.

As suggested by the reviewer, we have extracted the information about induction regimens and provided the results, and discussion in the revised manuscript. Because the recipients receiving anti-thymocyte globulin were significantly associated with 1- and 3-month mortalities in the multivariate analyses, the results of other variables such as age, coronary artery disease, cardiac arrhythmia, and hemodialysis in the early TRM, and age, coronary artery disease, acute myocardial infarction, cardiac arrhythmia, hemodialysis, and cytomegalovirus and fungal infections in TRM were also revised throughout the manuscript. In particular, the contents for CMV infection related to TRM including Fig 2B were eliminated because the P value was changed from 0.012 to 0.106 in the revised manuscript.

Abstract section (page 3, lines 8 to 15)

“Based on a multivariate analysis, older age (hazard ratio [HR] = 1.06; P < 0.001), coronary artery disease (HR = 3.02; P = 0.002), and hemodialysis compared with pre-emptive kidney transplantation (HR = 2.53; P = 0.046) were the risk factors for early TRM. Older age (HR = 1.07; P < 0.001), coronary artery disease (HR = 2.88; P < 0.001), and hemodialysis (HR = 2.35; P = 0.004) were the common independent risk factors for TRM. In contrast, cardiac arrhythmia (HR = 1.98; P = 0.027) was associated only with early TRM, and fungal infection (HR = 2.61; P < 0.001), and epoch of transplantation (HR = 0.34; P < 0.001) were the factors associated with only TRM.”

Materials and Methods section (page 6, line 15)

“The induction regimens such as basiliximab, and anti-thymocyte globulin were also extracted.”

Results section (page 8, lines 16 to 17)

“Regarding to induction therapy, basiliximab, and anti-thymocyte globulin were administered to 79.0%, and 11.4% of recipients, respectively.”

Results section (page 11, lines 16 to 18)

“Patients with anti-thymocyte globulin showed significant relation to TRM (P < 0.001), whereas those with basiliximab did not.”

Results section (page 11, lines 22 to page 12, lines 5)

“The risk factors of early TRM and TRM are shown in Tables 2 and 3, respectively. Based on the Cox multivariate analysis, older age (hazard ratio [HR] = 1.06; P < 0.001), CAD (HR = 3.02; P = 0.002), cardiac arrhythmia (HR = 1.98; P = 0.027), and hemodialysis compared to pre-emptive kidney transplant (HR = 2.53; P = 0.046) were independently associated with early TRM. Moreover, older age (HR = 1.07; P < 0.001), CAD (HR = 2.88, P = < 0.001), and hemodialysis (HR = 2.35, P = 0.004) were consistently independent risk factors of TRM at any time. However, fungal infection, (HR = 2.61; P < 0.001), and the epoch of transplantation (HR = 0.34 for 2010–2016; P < 0.001) were correlated to TRM only.”

Results section (page 15, lines 10 to 11)

“Fungal infection (Fig 2B) affected TRM (after early TRM). The protective effect of transplantation in 2010–2016 is illustrated in Fig 2C.”

Discussion section (page 17, lines 8 to 19)

“The use of anti-thymocyte globulin has been greater in high-risk recipients such as highly sensitized patients, recipients from deceased donors, re-transplantations, and ABO incompatible transplants [36]. According to a prospective, randomized study, patients receiving anti-thymocyte globulin presented a greater incidence of infection (85.8%) compared to those with basiliximab (75.2%) at 12 months after transplantation [37]. However, there was no significant difference in patient survival, similar to the results of a recent study using a network meta-analysis [38]. In Korea, the one-year patient survival in the anti-thymocyte globulin group (89.4%) was compared to the basiliximab group (93.8%), and presented no significant difference [39]. Based on our data, the high-risk recipients receiving anti-thymocyte globulin were significantly associated with early mortality. Further studies for the premature mortality are necessary to validate our results, and intensive care for the high-risk patients receiving anti-thymocyte globulin is important for improving outcomes.”

Figure (page 15, lines 18 to 23)

Fig 2. Cumulative incidence of mortality according to the factors associated 1- or 3-month mortality after kidney transplantation. 

(A) Cardiac arrhythmia was related to a worse outcome 1 month after transplantation. (B) Fungal infection were a risk factor of 3-month mortality after transplantation. (C) Recent epoch of transplantation (2010–2016) was a protective factor of 3-month mortality compared to the treatment-related mortality of previous epoch (2003–2009). 

Table 1. Comparison of the characteristics between living kidney recipients versus deceased ones at 1 and 3 months after transplantation.

Characteristicsa Early TRM TRM

 Living at 1 month Death by 1 month P-valueb Living at 3 months Death by 3 months P-valueb

Number (%) 15,999 74 　 15,913 160 　

Age, years 47 (37-55) 56 (48.8-61) < 0.001 47 (37-55) 55.5 (48-61) < 0.001

< 50 9,188 (57.4) 19 (25.7) < 0.001 9,163 (57.6) 44 (27.5) < 0.001

50–59 4,830 (30.2) 32 (43.2) 　 4,792 (30.1) 70 (43.8) 　

60–69 1,838 (11.5) 18 (24.3) 　 1,819 (11.4) 37 (23.1) 　

70–79 143 (0.9) 5 (6.8) 　 139 (0.9) 9 (5.6) 　

Sex, male 9,451 (59.1) 44 (59.5) 0.946 9,403 (59.1) 92 (57.5) 0.684 

Cause of ESRD 

Diabetes mellitus 3,501 (21.9) 19 (25.7) 0.431 3,479 (21.9) 41 (25.6) 0.252

Hypertension 2,001 (12.5) 8 (10.8) 0.660 1,992 (12.5) 17 (10.6) 0.471

Glomerulonephritis 2,850 (17.8) 11 (14.9) 0.508 2,840 (17.8) 21 (13.1) 0.120

Cystic kidney disease 368 (2.3) 2 (2.7) 0.818 365 (2.3) 5 (3.1) 0.485

Underlying diseasec 　 　 　 　 　 　

 Cardiac disease 　 　 　 　 　 　

 Coronary artery disease 392 (2.5) 9 (12.2) < 0.001 384 (2.4) 17 (10.6) < 0.001

 Acute myocardial infarction 288 (1.8) 2 (2.7) 0.561 283 (1.8) 7 (4.4) 0.014 

 Cardiac arrhythmia 1,240 (7.8) 13 (17.6) 0.002 1,230 (7.7) 23 (14.4) 0.002 

 Cerebrovascular disease 　 　 　 　 　 　

 Cerebral hemorrhage 54 (0.3) 0 (0.0) 0.617 54 (0.3) 0 (0.0) 0.460 

 Cerebral infarction 247 (1.5) 1 (1.4) 0.893 246 (1.5) 2 (1.3) 0.763 

Hemodialysis 13,134 (82.1) 69 (93.2) 0.012 13,055 (82.0) 148 (92.5) 0.001 

Dialysis vintage, monthsd 42.5 (29.5-62.8) 16.0 (9.5-24.5) 0.051 41.0 (29.0-63.5) 24.5 (12.8-39.0) 0.179

Before steroid usee 1,148 (7.2) 3 (4.1) 0.416 1,140 (7.2) 11 (6.9) 1.000

Induction therapy 

Basiliximab 12,637 (79.0) 55 (74.3) 0.402 12,569 (79.0) 123 (76.9) 0.579

Anti-thymocyte globulin 1,818 (11.4) 22 (29.7) < 0.001 1,799 (11.3) 41 (25.6) < 0.001

Infection 　 　 　 　 　 　

 CMV infection 694 (4.3) 4 (5.4) 0.653 1,900 (11.9) 37 (23.1) < 0.001

 Fungal infection 639 (4.0) 2 (2.7) 0.571 1,205 (7.6) 37 (23.1) < 0.001

Epoch of transplantation 　 　 　 　 　 　

 2003–2009 4,634 (29.0) 27 (36.5) 0.155 4,594 (28.9) 67 (41.9) < 0.001

 2010–2016 11,365 (71.0) 47 (63.5) 　 11,319 (71.1) 93 (58.1) 　

a Data were expressed as number (%) or median (interquartile range).

b P value was calculated using chi-square test or Mann–Whitney U test.

c In case of the presence of underlying diseases, multiple diseases were designated to one patient.

d Data were obtained from the Korean Network for Organ Sharing system.

e The use of intravenous steroids such as dexamethasone, and prednisolone within 6 months before transplantation.

Abbreviations: CMV, cytomegalovirus; ESRD, end-stage renal disease; TRM, treatment-related mortality.

Table 2. Univariate and multivariate analyses of 1-month mortality after kidney transplantation.

Variable Univariate Multivariate

 HR (95% CI) P-value HR (95% CI) P-value

Age, yearsa 1.07 (1.05–1.10) < 0.001 1.06 (1.04-1.09) < 0.001

 < 50 Reference 　 　 　

 50–59 3.21 (1.82-5.66) < 0.001 　 　

 60–69 4.74 (2.49-9.03) < 0.001 　 　

 70–79 16.66 (6.22-44.62) < 0.001 　 　

Sex, male 0.98 (0.62-1.56) 0.944 　 　

Cause of ESRD 

 Diabetes mellitus 1.22 (0.89-1.75) 0.451 

 Hypertension 0.80 (0.58-1.12) 0.672 

 Glomerulonephritis 0.93 (0.52-2.15) 0.591 

 Cystic kidney disease 1.19 (0.78-2.32) 0.854 

Underlying disease 　 　 　 　

 Cardiac disease 　 　 　 　

 Coronary artery disease 5.51 (2.74-11.06) < 0.001 3.02 (1.48-6.17) 0.002 

 Acute myocardial infarction 1.51 (0.37-6.15) 0.566 　 　

 Cardiac arrhythmia 2.53 (1.39-4.60) 0.002 1.98 (1.08-3.62) 0.027 

 Cerebrovascular disease 　 　 　 　

 Cerebral hemorrhage NA 　 　 　

 Cerebral infarction 0.87 (0.12-6.26) 0.890 　 　

Hemodialysis 3.00 (1.21-7.45) 0.018 2.53 (1.02-6.28) 0.046 

Dialysis vintage, monthsc 0.918 (0.833-1.012) 0.086 

Before steroid used 0.55 (0.17-1.74) 0.307 

Induction therapy 

Basiliximab 0.77 (0.46-1.30) 0.326 

Anti-thymocyte globulin 3.31 (2.01-5.45) < 0.001 2.62 (1.59-4.32) < 0.001

Infection 　 　 　 　

 CMV infection 1.26 (0.46-3.45) 0.652 　 　

 Fungal infection 0.66 (0.16-2.71) 0.569 　 　

Epoch of transplantation, 2010–2016 0.72 (0.45-1.15) 0.168 　 　

a Variables less than 0.05 of P-values in univariate analysis were included in the multivariate analysis.

b NA is presented if the paucity of deceased or living patients exists for each variable 1 month after kidney transplantation.

c Data were obtained from the Korean Network for Organ Sharing system.

d The use of intravenous steroids such as dexamethasone, and prednisolone within 6 months before transplantation.

Abbreviations: CI, confidence interval; CMV, cytomegalovirus; ESRD, end-stage renal disease; HR, hazard ratio; NA, not applicable.

Table 3. Univariate and multivariate analyses of 3-month mortality after kidney transplantation.

Variable Univariate Multivariate

 HR (95% CI) P-value HR (95% CI) P-value

Age, yearsa 1.07 (1.05–1.09) < 0.001 1.07 (1.05-1.09) < 0.001

 < 50 　 　 　 　

 50–59 3.05 (2.09-4.44) < 0.001 　 　

 60–69 4.24 (2.74-6.57) < 0.001 　 　

 70–79 13.16 (6.43-26.96) < 0.001 　 　

Sex, female 1.07 (0.78-1.46) 0.690 　 　

Cause of ESRD 

 Diabetes mellitus 1.25 (0.92-1.59) 0.273 

 Hypertension 0.86 (0.68-1.10) 0.463 

 Glomerulonephritis 0.91 (0.49-1.75) 0.385 

 Cystic kidney disease 1.23 (0.87-2.41) 0.526 

Underlying disease 　 　 　 　

 Cardiac disease 　 　 　 　

 Coronary artery disease 4.82 (2.92-7.97) < 0.001 2.88 (1.71-4.84) < 0.001 

 Acute myocardial infarction 2.48 (1.16-5.29) 0.019 1.75 (0.81-3.80) 0.157 

 Cardiac arrhythmia 1.99 (1.28-3.10) 0.002 1.40 (0.89-2.18) 0.145 

 Cerebrovascular disease 　 　 　 　

 Cerebral hemorrhage NA 　 　 　

 Cerebral infarction 0.80 (0.20-3.23) 0.755 　 　

Hemodialysis 2.69 (1.49-4.85) 0.001 2.35 (1.30-4.25) 0.004 

Dialysis vintage, monthsc 0.963 (0.911-1.017) 0.179 

Before steroid used 0.95 (0.52-1.76) 0.882 

Induction therapy 

Basiliximab 0.88 (0.61-1.28) 0.514 

Anti-thymocyte globulin 2.73 (1.92-3.90) < 0.001 2.38 (1.62-3.49) < 0.001

Infection 　 　 　 　

 CMV infection 2.19 (1.51-3.16) < 0.001 1.39 (0.93-2.08) 0.106 

 Fungal infection 3.57 (2.47-5.15) < 0.001 2.61 (1.79-3.82) < 0.001

Epoch of transplantation, 2010–2016 0.58 (0.42-0.79) 0.001 0.34 (0.24-0.48) < 0.001

a Variables less than 0.05 of P-values in univariate analysis were included in the multivariate analysis.

b NA is presented if the paucity of deceased or living patients exists for each variable 3 months after kidney transplantation.

c Data were obtained from the Korean Network for Organ Sharing system.

d The use of intravenous steroids such as dexamethasone, and prednisolone within 6 months before transplantation.

Abbreviations: CI, confidence interval; CMV, cytomegalovirus; ESRD, end-stage renal disease; HR, hazard ratio; NA, not applicable.

5. Results. A sub-analysis might be performed distinguishing TRM analysis between transplantation cases performed in different epoch (2003-2009 vs 2010-2016). This might be interesting even if this factor did not reach statistical significance at univariate analysis. In fact, “Epoch of transplantation 2010–2016” showed a trend towards statistical significance.

We have conducted the sub-analysis for the two epoch of transplantation and added the results to the revised Results and Discussion sections as follows. 

Results section (page 12, lines 6 to 10)

“Regarding to the epoch of transplantation, the aged between 50 and 59 years (HR = 0.37, P = 0.005 for early TRM; HR = 0.37, P < 0.001 for TRM), the patients receiving basiliximab as induction therapy (HR = 0.44, P = 0.002 for early TRM; HR = 0.40, P < 0.001 for TRM), and recipients with CMV infection (HR = 0.13, P = 0.040 for early TRM; HR = 0.39, P = 0.005 for TRM) presented better outcome in 2010-2016, when compared to 2003-2009.”

Discussion section (page 18, lines 22 to page 19, lines 4)

“In particular, relatively low- or intermediate-risk patients such as aged 50 to 59 years, and patients receiving basiliximab were influence by the improved protocols, and showed better outcome than high-risk recipients (aged over 60 years, and recipients with anti-thymocyte globulin). Further, more aggressive and sophisticated infection controls on CMV such as monitoring quantitative levels, and high dose of antiviral therapy [47] may protect more recipients in 2010-2016 than those in 2003-2009.”

6. Results. In the following sentence, Authors should specify the number of recipients who died among the total number of recipients for each type of transplantation, and only after the percentage value in parentheses: “The rates of recipients who died 1 month (1.2% for living, 2.4% for deceased, and

3.3% for non-heart beating) and 3 months (3.2% for living, 5.5% for deceased, and 3.3% for non-heart beating) after transplantation showed significant difference according to the donor state (P < 0.001).”

We have applied these results to our cohort and presented the number of recipients in the revised Results section (page 11, lines 9 to 12) as follows. The calculation errors were also corrected.

“The rates of recipients who died 1 month (n = 1, 1.4% for living; n= 2, 2.7% for deceased; and n = 2, 2.7% for non-heart beating) and 3 months (n = 5, 3.1% for living; n = 9, 5.6% for deceased; and n = 5, 3.1% for non-heart beating) after transplantation showed significant difference according to the donor state (P < 0.001).”

Response to the editor’s comment

We have uploaded our revised figure files (Fig 1.tif, and Fig 2.tif) to the PACE digital diagnostic tool to meet PLOS requirements. The preview files (Preview_20200623002121848.pdf, and Preview_20200623002149051.pdf) were generated and checked.

---

## [Decision Letter · Decision Letter 2]

6 Jul 2020

The risk factors associated with treatment-related mortality in 16,073 kidney transplantation - A nationwide cohort study

PONE-D-20-03919R2

Dear Dr. Jeong,

We’re pleased to inform you that your manuscript has been judged scientifically suitable for publication and will be formally accepted for publication once it meets all outstanding technical requirements.

Kind regards,

Academic Editor

PLOS ONE

Additional Editor Comments (optional):

Reviewers' comments:

Reviewer's Responses to Questions

**Comments to the Author**

1. If the authors have adequately addressed your comments raised in a previous round of review and you feel that this manuscript is now acceptable for publication, you may indicate that here to bypass the “Comments to the Author” section, enter your conflict of interest statement in the “Confidential to Editor” section, and submit your "Accept" recommendation.

Reviewer #1: All comments have been addressed

Reviewer #2: All comments have been addressed

2. Is the manuscript technically sound, and do the data support the conclusions?

Reviewer #1: Yes

Reviewer #2: Yes

3. Has the statistical analysis been performed appropriately and rigorously? 

Reviewer #1: Yes

Reviewer #2: Yes

4. Have the authors made all data underlying the findings in their manuscript fully available?

Reviewer #1: Yes

Reviewer #2: Yes

5. Is the manuscript presented in an intelligible fashion and written in standard English?

Reviewer #1: Yes

Reviewer #2: Yes

6. Review Comments to the Author

Reviewer #1: I thank the authors to have addressed all my recommendations. At the moment the

paper is suitable for publication

Reviewer #2: Treatment-related mortality (TRM) after renal transplantation is a concept different from disease-related mortality and appeared to be a very prevalent entity.

Strengths:

- There are scarce data in scientific literature about TRM within 1 or 3 months after kidney transplantation.

- Authors collected an important amount of data from a very large cohort of patients using a national population based database, which included information about a total of 16,073 kidney recipients.

This article would be a valuable contribution to the medical literature to encourage further discussion on this entity.

The writing is clear and easily understandable.

The Authors have worked hard to improve this article, and they have met all criticisms raised by referees.

I feel the manuscript is now suitable for publication in Plos One.

7. PLOS authors have the option to publish the peer review history of their article (what does this mean?). If published, this will include your full peer review and any attached files.

Reviewer #1: No

Reviewer #2: No

---

## [Editor Report · Acceptance letter]

9 Jul 2020

PONE-D-20-03919R2 

The risk factors associated with treatment-related mortality in 16,073 kidney transplantation - A nationwide cohort study 

Dear Dr. Jeong:

I'm pleased to inform you that your manuscript has been deemed suitable for publication in PLOS ONE. Congratulations! Your manuscript is now with our production department. 

Kind regards, 

on behalf of

Dr. Robert Jeenchen Chen 

Academic Editor

PLOS ONE